# *λ-ECLIPSE*: Multi-Concept Personalized Text-to-Image Diffusion Models by Leveraging CLIP Latent Space

**Maitreya Patel**[*]                                   *maitreya.patel@asu.edu*
*Arizona State University*

**Sangmin Jung**[*]                                   *sgmin.jung@asu.edu*
*Arizona State University*

**Chitta Baral**                                   *chitta@asu.edu*
*Arizona State University*

**Yezhou Yang**                                   *yz.yang@asu.edu*
*Arizona State University*

**Reviewed on OpenReview:** *https://openreview.net/forum?id=7q5UewlAdM*

## Abstract

Despite the recent advances in personalized text-to-image (P-T2I) generative models, it remains challenging to perform finetuning-free multi-subject-driven T2I in a resource-efficient manner. Predominantly, contemporary approaches, involving the training of hypernetworks and Multimodal Large Language Models (MLLMs), require heavy computing resources that range from 600 to 12300 GPU hours of training. These subject-driven T2I methods hinge on Latent Diffusion Models (LDMs), which facilitate T2I mapping through cross-attention layers. While LDMs offer distinct advantages, P-T2I methods' reliance on the latent space of these diffusion models significantly escalates resource demands, leading to inconsistent results and necessitating numerous iterations for a single desired image.

Through empirical evidences we find that CLIP (vision) latent space is already expressive enough to preserve the fine-grained details. Building upon this insight, in this paper, we present *λ-ECLIPSE*, a diffusion-agnostic prior-training strategy that operates in the latent space of a pre-trained CLIP model without relying on the diffusion UNet models. *λ-ECLIPSE* leverages the image-text interleaved pre-training for fast and effective multi-subject-driven P-T2I. Through extensive experiments, we establish that *λ-ECLIPSE* surpasses existing baselines in composition alignment while preserving concept alignment performance, even with significantly lower resource utilization. *λ-ECLIPSE* performs multi-subject driven P-T2I with just 34M parameters and is trained on a mere 74 GPU hours. Additionally, *λ-ECLIPSE* demonstrates the unique ability to perform multi-concept interpolations. Project page: `https://eclipse-t2i.github.io/Lambda-ECLIPSE/`

## 1 Introduction

The field of text-to-image (T2I) diffusion models has recently witnessed remarkable advancements, achieving greater photorealism and enhanced adherence to textual prompts. This has catalyzed the emergence of diverse applications, notably subject-driven personalized T2I (P-T2I) models. In particular, this encompasses the intricate task of learning and reproducing novel visual concepts or subjects in varied contexts requiring high concept and compositional alignment. The complexity escalates further when multi-subject personalization is desired.

---

[*]Equal contribution

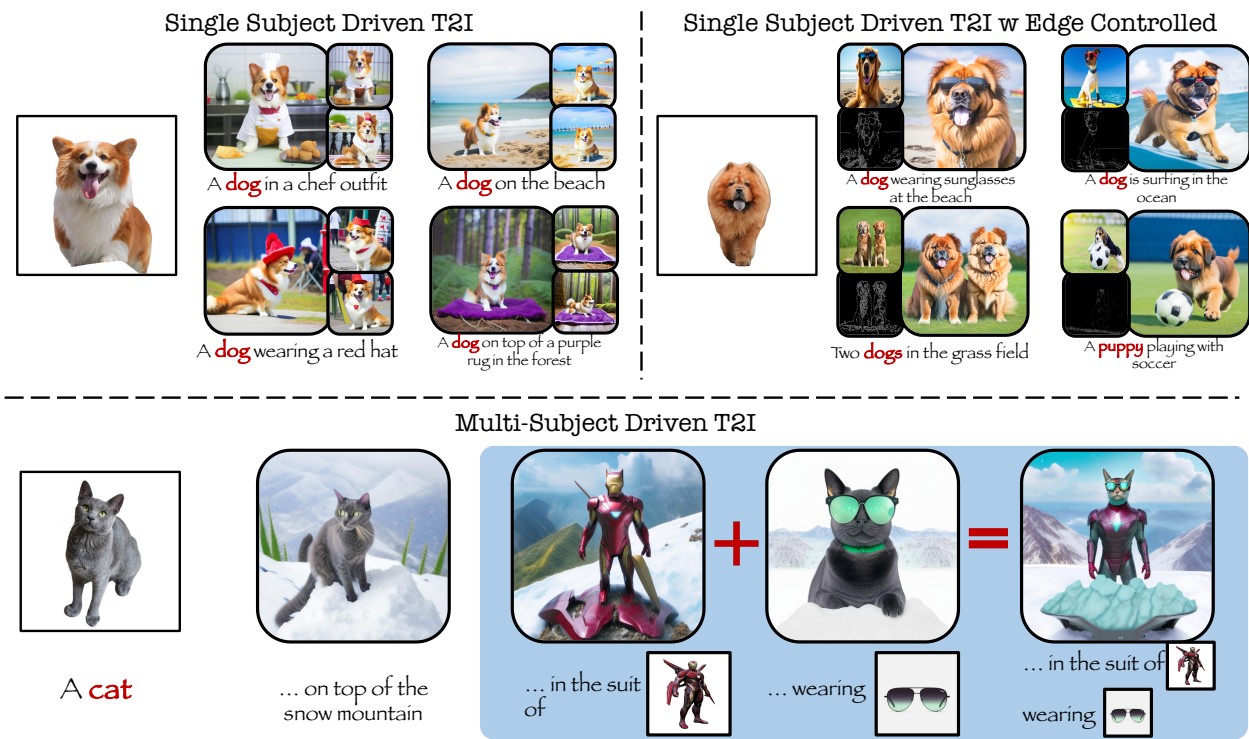

Figure 1: $\lambda$-`ECLIPSE` can estimate subject-specific latent image embeddings while maintaining the balance between concept and composition alignment in the CLIP latent space itself.

Early works employed concept-specific optimization strategies involving fine-tuning certain parameters within T2I diffusion models Gal et al. (2022); Ruiz et al. (2023a); Kumari et al. (2023); Tumanyan et al. (2023); Gal et al. (2023). Although these methods achieve state-of-the-art (SOTA) performance, they struggle with generalization and are time-intensive. Contemporary research is pivoting towards fast personalization techniques. Within this paradigm, there are two types of popular approaches: 1) Methods that involve training hypernetworks and integrating new layers or parameters within pre-trained diffusion UNet models Wei et al. (2023); Ye et al. (2023); Tewel et al. (2023); Shah et al. (2023); Ruiz et al. (2023b), and 2) MLLM-based learning of prior models that focuses on leveraging text-latent space of frozen diffusion UNet model Pan et al. (2023); Sun et al. (2023).

The hypernetwork-based strategy achieves single-concept customization but has not been extended to multi-concepts. Moreover, when combined with additional control (i.e. Canny edge map), they struggle to maintain the concept alignment (~30% drop in performance; Section 4) and strongly favor the edge map. At the same time, MLLM-based approaches can perform fast multi-concept customization but require heavy computing resources. In Table 1, we provide the overview of various single and multi-concept customization methodologies in terms of total parameters, iterations, and GPU hours required to train the models. It can be observed that multi-concept customization methodologies further increase the resource requirements. For instance, Kosmos-G Pan et al. (2023) consumes 18x more resources than IP-Adapter Ye et al. (2023). And Emu2 Sun et al. (2023) requires training of 19x more parameters compared to Kosmos-G. Hence, despite MLLMs' seemingly useful scenarios, it is not viable to blindly train them.

Upon further investigation, we find that most subject-driven T2I approaches build upon variants of the Latent Diffusion Model (LDM) Rombach et al. (2022), specifically Stable Diffusion models. These LDMs employ cross-attention layers to condition diffusion models with text embeddings, necessitating a mapping of target subject images to latent spaces compatible with the diffusion models at the prior training stage. This is also known as score distillation instruction tuning for MLLMs Pan et al. (2023). As there is no choice but to learn this text-to-image diffusion latent space, it involves backpropagation through the entire diffusion

Table 1: **A quick overview of previous works on P-T2I.** Our method is the first to offer multi-concept-driven generation without depending on diffusion UNet models (except for inference). We provide the extended overview table in the appendix.

| Method | Multi Concepts | Finetuning Free | Diffusion Free | Total opt. params | Training Steps | Dataset Size | GPU Hours |
|---|---|---|---|---|---|---|---|
| Textual Inversion Gal et al. (2022) | ✗ | ✗ | ✗ | 768 | 5000 | - | 1 |
| DreamBooth Ruiz et al. (2023a) | ✗ | ✗ | ✗ | 0.9B | 800 | - | 0.2 |
| ELITE Wei et al. (2023) | ✗ | ✓ | ✗ | 77M | 135K | 125K | 336 |
| BLIP-Diffusion Li et al. (2023a) | ✗ | ✓ | ✗ | 1.5B | 500K | 129M | 2304 |
| IP-Adapter Ye et al. (2023) | ✗ | ✓ | ✗ | 22M | 1M | - | 672 |
| Custom Diffusion Kumari et al. (2023) | ✓ | ✗ | ✗ | 57M | 500 | - | 0.1 |
| Subject-Diffusion Ma et al. (2023a) | ✓ | ✓ | ✗ | 252M | 300K | 76M | - |
| Kosmos-G Pan et al. (2023) | ✓ | ✓ | ✗ | 1.9B | 800K | 200M | 12300 |
| Emu-2 Sun et al. (2023) | ✓ | ✓ | ✗ | 37B | 70K | 162M | - |
| $\lambda$-*ECLIPSE* (ours) | ✓ | ✓ | ✓ | 34M | 100K | 2M | 74 |

model often comprising over a billion parameters, contributing to the inefficiency of existing P-T2I methods. Therefore, in this work, we focus on answering one question: `Do we really need diffusion models to train the customization models?`

To answer this question and improve the resource efficiency for multi-concept image generation, we present $\lambda$-*ECLIPSE*[1], which leverages the properties of UnCLIP T2I models (e.g. DALL-E 2 Ramesh et al. (2022) and Kandinsky v2.2 Razzhigaev et al. (2023)) and performs P-T2I in the compressed latent space of frozen CLIP model. Specifically, unlike previous MLLM-based methodologies, $\lambda$-*ECLIPSE* aligns the output space of priors with CLIP vision space instead of the CLIP text space. $\lambda$-*ECLIPSE* takes multiple images and text instructions as input and estimates the respective vision embeddings, which can be used by the frozen diffusion UNet model from the UnCLIP stack to generate the resulting image. This elevates the training time dependencies on diffusion models for P-T2I; significantly contributing to the resource efficiency. Additionally, as diffusion or MLLM-based priors are still compute heavy due to a huge number of parameters and slower convergence, we build upon *ECLIPSE* Patel et al. (2023b) and SEED Ge et al. (2023), which shows that text-to-image mapping can be optimized through contrastive pre-training. Here, we select *ECLIPSE* as preferred choice of prior architecture for best efficiency. At last, we propose a subject-driven instruction tuning task based on the image-text interleaved data as a pre-training strategy. This involves creating 2 million high-quality image-text pairs, where text embeddings linked to subjects are substituted with the respective image embeddings, which in return are considered as input to the $\lambda$-*ECLIPSE*. While $\lambda$-*ECLIPSE* can be plugged with these pre-trained methods, we explore the possibility of $\lambda$-*ECLIPSE* to incorporate Canny edge as an additional coarse-level control to synergetically work with subject-driven image generation tasks. Figure 1 provides the overview of $\lambda$-*ECLIPSE* capabilities.

Overall, we propose $\lambda$-*ECLIPSE* as an initial attempt to motivate future works on designing resource-efficient solutions for MLLM-based approaches. We summarize our main contributions as follows:

- We show that P-T2I mapping can be learned in CLIP latent space without training-time dependency on the diffusion models; enabling efficient training and fast multi-subject customization.
- Extensive experiments on Dreambench, Multibench, and ConceptBed reveal that $\lambda$-*ECLIPSE* (34M parameter model) trained on a mere 74 GPU hours can achieve competitive performance to that of big counterparts (having 2B-37B parameters) and improve text-composition alignment.
- At last, $\lambda$-*ECLIPSE* inherits the smooth CLIP latent space. This allows us to perform the seamless transition between multi-concept generated images.

---

[1]The designation $\lambda$-*ECLIPSE* is inspired by its conceptual alignment with the $\lambda$-calculus. In this context, the $\lambda$-*ECLIPSE* model functions similarly to a functional abstraction within $\lambda$-calculus, where it effectively binds variables. These variables, in our case, represent novel visual concepts that are integrated through composition prompts. Here, *ECLIPSE* indicates our architecture design choice.

## 2 Related Works

**Text-to-Image Generative Models.** Pioneering efforts in image generation, notably DALL-E Ramesh et al. (2021) and CogView Ding et al. (2021), leveraged autoregressive models to achieve significant results. Recent advancements predominantly feature diffusion models, acclaimed for their high image fidelity and diversity in text-to-image (T2I) generation. A notable example is Stable Diffusion, which builds upon the Latent Diffusion Model (LDM) Rombach et al. (2022) and excels in semantic and conceptual understanding by transitioning training to latent space. Imagen Saharia et al. (2022), Pixart-$\alpha$ Chen et al. (2023b), and DALL-E 3 Betker et al. (2023) propose using a large T5 language model to improve language understanding. DALL-E 2 Ramesh et al. (2022) along with its UnCLIP variation models such as Kandinsky Razzhigaev et al. (2023) and Karlo Lee et al. (2022), uses a diffusion prior and diffusion UNet modules to generate images using the pre-trained CLIP Radford et al. (2021) model.

**Personalized T2I Methods.** Approaches like Textual Inversion Gal et al. (2022), DreamBooth Ruiz et al. (2023a), and Custom Diffusion Kumari et al. (2023) focus on training specific parameters to encapsulate visual concepts. LoRA Hu et al. (2021) and Perfusion Tewel et al. (2023) target efficient fine-tuning adjustments, particularly rank 1 modifications. However, these methods are constrained by their requirement for concept-specific tuning. ELITE Wei et al. (2023) was the first approach addressing fast customized generation for single-subject T2I. BLIP-Diffusion Li et al. (2023a) adapts the BLIP2 encoder Li et al. (2023b), training approximately 1.5B parameters to enable zero-shot, subject-driven image generation. IP-Adapter introduces a decoupled cross-attention mechanism, negating the need to train the foundational UNet model by permitting fine-tuning of a reduced number of 22M parameters.

Mix-of-Show Gu et al. (2023) and Zip-LoRA Shah et al. (2023) train individual concepts and then combine them to generate multiple subjects. Break-A-Scene Avrahami et al. (2023) shows multi-concept capability but requires single images containing diverse objects. Subject Diffusion Ma et al. (2023a) creates a high-quality dataset and presents the precision control for fast personalized multi-subject image generation. Kosmos-G and Emu2 Sun et al. (2023), akin to Subject-Diffusion Ma et al. (2023a), employs a Multimodal Large Language Model (MLLM) for text-image embedding alignment, though it necessitates extensive parameter optimization (1.9B-37B). These multi-subject P-T2I methods are not only demanding in terms of parameters but also depend on a massive number of frozen parameters of the diffusion UNet model, increasing training computational loads. In contrast, our model, $\lambda$-*ECLIPSE*, forgoes test-time fine-tuning and training-time reliance on the diffusion UNet model for single and multi-concept, control-guided P-T2I, positioning it as a resource-efficient solution.

At last, methods like GLIGEN Li et al. (2023c), ControlNet Zhang et al. (2023a), and UniControl Qin et al. (2023) incorporate additional controls (i.e., edge map, depth, segmentations) into the diffusion model to generate the desired images. BLIP-Diffusion, IP-Adapter, and Kosmos-G can leverage such pre-trained controls. However, in many scenarios, these controls are too strong, making generated images lose subject-specific details. We show that $\lambda$-*ECLIPSE* learns to balance the edge map, subjects, and composition. We offer a more comprehensive review of related works in the appendix.

## 3 Method

In this section, we introduce $\lambda$-*ECLIPSE*, our approach to multi-subject personalized text-to-image generation. Our method combines the contrastive text-to-image strategy from *ECLIPSE* with the novel image-text interleaved pretraining strategy, notably omitting the need for explicit diffusion modeling. Our approach mainly capitalizes on the efficient utilization of the CLIP latent space. Figure 2 outlines the end-to-end framework.

The primary objective of $\lambda$-*ECLIPSE* is to facilitate single and multi-subject P-T2I generation processes, accommodating edge maps as conditional guidance. Initially, we detail the problem formulation and elaborate on the UnCLIP stack design of the $\lambda$-*ECLIPSE*. Subsequently, we delve into the image-text interleaved training methodology. This fine-tuning process enables the $\lambda$-*ECLIPSE* to harness semantic correlations between CLIP image and text latent spaces while preserving subject-specific visual features.

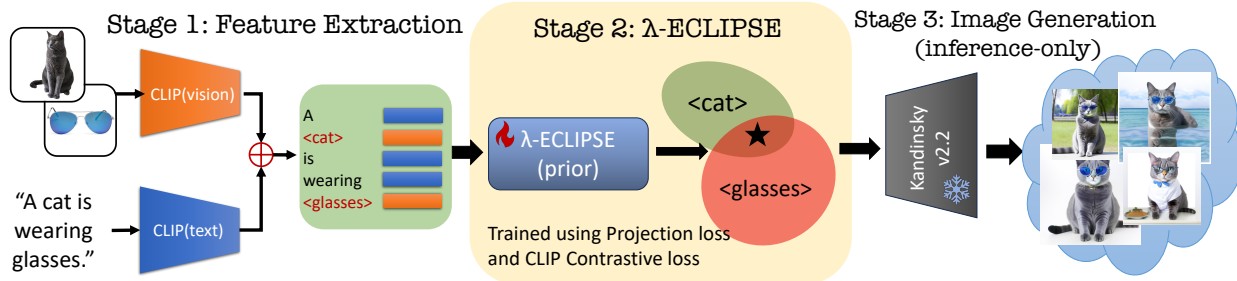

Figure 2: **Three stages of the λ-*ECLIPSE* pipeline.** 1) Create the image-text interleaved features using frozen CLIP. 2) Pre-train the λ-*ECLIPSE* (34M parameters) using Eq. 1, which ensures the mapping to the desired latent space given the image-text interleaved data. 3) During inference, the frozen Kandinsky v2.2 diffusion UNet model takes the output from the λ-*ECLIPSE* and generates the image.

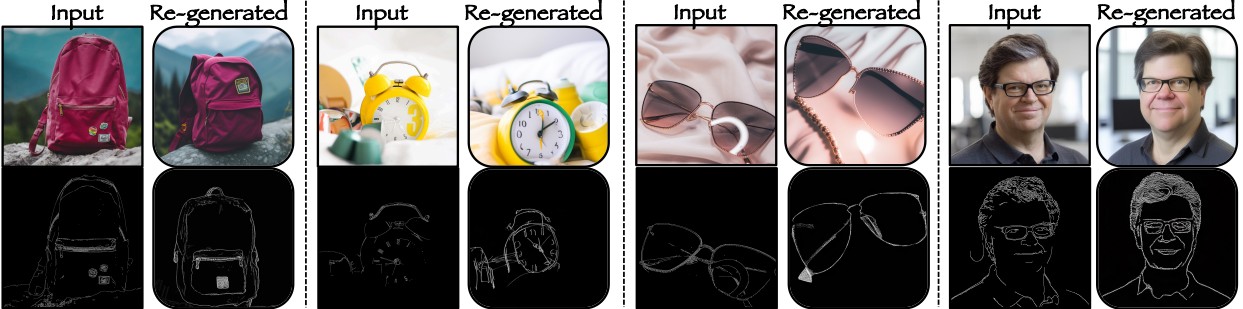

Figure 3: **CLIP(vision) features capture the semantics and fine-grained visual details.** Each input is given as input to the Kandinsky v2.2 and re-generated from the decoder. (Top: Real-images, Bottom: Canny edge)

### 3.1 Text-to-Image Prior Mapping

In the UnCLIP T2I models, the objective of the text-to-image prior model ($f_\theta$) is to establish a proficient text-to-image embedding mapping. This model is designed to adeptly map textual representations to their corresponding visual embeddings, denoted as ($f_\theta : z_y \rightarrow z_x$), where $z_{x/y}$ represent the embeddings for images and text, respectively. The visual embedding predictions ($\hat{z}_x = f_\theta(z_y)$) are then effectively utilized by the diffusion image generators ($h_\phi$), which are inherently conditioned on these vision embeddings ($h_\phi : z_x \rightarrow x$). In our experiments, we utilize the Kandinsky v2.2 diffusion UNet model as $h_\phi$.

As shown in Figure 3, the CLIP vision encoder is very expressive and preserves the finegrained details in $z_x$ that is required to reconstruct the input image. CLIP image embedding itself achieves the high concept alignment score (`DINO: 0.66`) similar to the finetuning-based DreamBooth method Ruiz et al. (2023a).

Our goal is to accurately estimate the image embedding $\hat{z}_x$, incorporating the subject representations, thereby eliminating reliance on $h_\phi$ during training. Existing LDM-based P-T2I methods are limited by the LDM's singular module approach ($h_\phi : z_y \rightarrow x$). Consequently, mastering the latent space of $h_\phi$ becomes essential for effective P-T2I for the baseline methodologies, which limits the previous methodologies.

We propose a new mapping function, $f_\theta$, which processes text representations ($z_y$) alongside subject ($x_k$) specific visual representations ($z_{x_k}$), to derive an image embedding that encapsulates both text prompts and subject visuals ($\hat{z}_x$). The challenge lies in harmonizing $z_{x_k}$ and $z_y$ within $f_\theta : (z_y, z_{x_k}) \rightarrow \hat{z}_x$, ensuring alignment while preventing overemphasis on either aspect, as this could compromise composition alignment. To address this, we employ the contrastive pre-training strategy after Patel et al. (2023b):

$$\mathcal{L}_{prior} = \mathop{\mathbb{E}}_{\substack{\epsilon \sim \mathcal{N}(0,I) \\ z_y, z_{x_k}}} \Big[||z_x - f_\theta(\epsilon, z_y, z_{x_k})||_2^2\Big] - \frac{\lambda}{N} \sum_{i=0}^{N} \log \frac{\exp(\langle \hat{z}_x^i, z_y^i \rangle / \tau)}{\sum_{j \in [N]} \exp(\langle \hat{z}_x^i, z_y^j \rangle / \tau)}. \tag{1}$$

Here, $\lambda$ serves as the hyperparameter. $i$ and $j$ represent the index of the given input batch with the size $N$. $\langle \cdot \rangle$ represents the inner-product and $\tau$ is the temperature parameter. The first loss term (projection loss) measures the mean-squared error between the estimated and actual image embeddings, primarily ensuring concept alignment. However, our preliminary studies reveal that exclusive reliance on this term diminishes composition alignment. Therefore, we stick with the contrastive loss component (the second loss term) to bolster compositional generalization, with $\lambda$ balancing concept and composition alignment.

**Additional Coarse-level Control-based T2I Prior Mapping.** Acknowledging the limitations in existing methods, which necessitate learning the diffusion latent space even for additional control inputs, we endeavor to achieve a more nuanced balance between subject, text, and supplementary conditions. Consequently, we have augmented $\lambda$-*ECLIPSE* to accommodate an additional modality, a Canny edge map, providing more refined control over subject-driven image generation. This entails modifying the prior model to accept additional conditions ($f'_\theta : (z_y, z_{x_i}, z_c) \to \hat{z}_x$, here $z_c$ is the additional modality embedding).

Additionally, during training, we drop $z_c$ for 1% to improve the image generations without relying on the edge map. This enhances the stability and broadens the generalization capabilities of $\lambda$-*ECLIPSE*, yielding benefits even in the absence of these controls during inference. Our results demonstrate that $\lambda$-*ECLIPSE* learns a unified mapping function, accurately estimating target image representations through the effective integration of text, image, and edge maps – leading to learning coarse-level controls instead of hard constraints.

### 3.2 Image-text Interleaved Training

Our approach targets developing a versatile prior model capable of processing diverse inputs to estimate target visual outputs. Drawing from earlier methodologies, a straightforward solution involves concatenating different inputs, like combining text ("a dog wearing sunglasses") with respective concept-specific images. Preliminary experiments indicated that this method does not effectively capture the intricate relationships between target text tokens (e.g. "dog") and the corresponding concept images, especially when multiple concepts are present.

To address this, we adopt the interleaved pre-training strategy used in Kosmos-G, but with a notable modification to enhance resource efficiency. We incorporate pretrained frozen CLIP text and vision encoders for extracting modality-specific embeddings—separating text-only from subject-specific images. The key refinement in our process is the substitution of subject token-specific text embeddings with corresponding vision embeddings instead of introducing additional trainable tokens to handle the image embeddings via resampler Alayrac et al. (2022). First, we extract reference concept visual features ($z_{x_k} \in \mathcal{R}^{1 \times 1280}$) from the CLIP vision encoder. Similarly, we also extract the text prompt features ($z_y \in \mathcal{R}^{77 \times 1280}$) from the last layer of the CLIP text encoder. Here, 1280 is the CLIP-specific feature dimension. At last, we replace the concept noun corresponding latent features from $z_y$ with $z_{x_k}$; resulting in image-text interleaved features while preserving the contextual information of the text features. This alteration allows us to bypass the need to train the big priors models (e.g. MLLMs), significantly improving the model's proficiency in handling interleaved data.

For the generation of high-quality training datasets, we carefully selected 2 million high-quality images from the LAION dataset Schuhmann et al. (2022), each with a resolution of 1024x1024. Utilizing BLIP2, we generate captions for these images and employ SAM Kirillov et al. (2023) for extracting noun or subject-specific segmentation masks. Given the CLIP model's requirement for 224x224 resolution images, we avoid resizing the masks within their original resolutions. Instead, we opt for cropping the area of interest using Grounding DINO Liu et al. (2023a), followed by resizing the masked object while preserving its aspect ratio. This technique is crucial in retaining maximum visual information for each subject during the training phase. We provide more details about the filters used in the appendix.

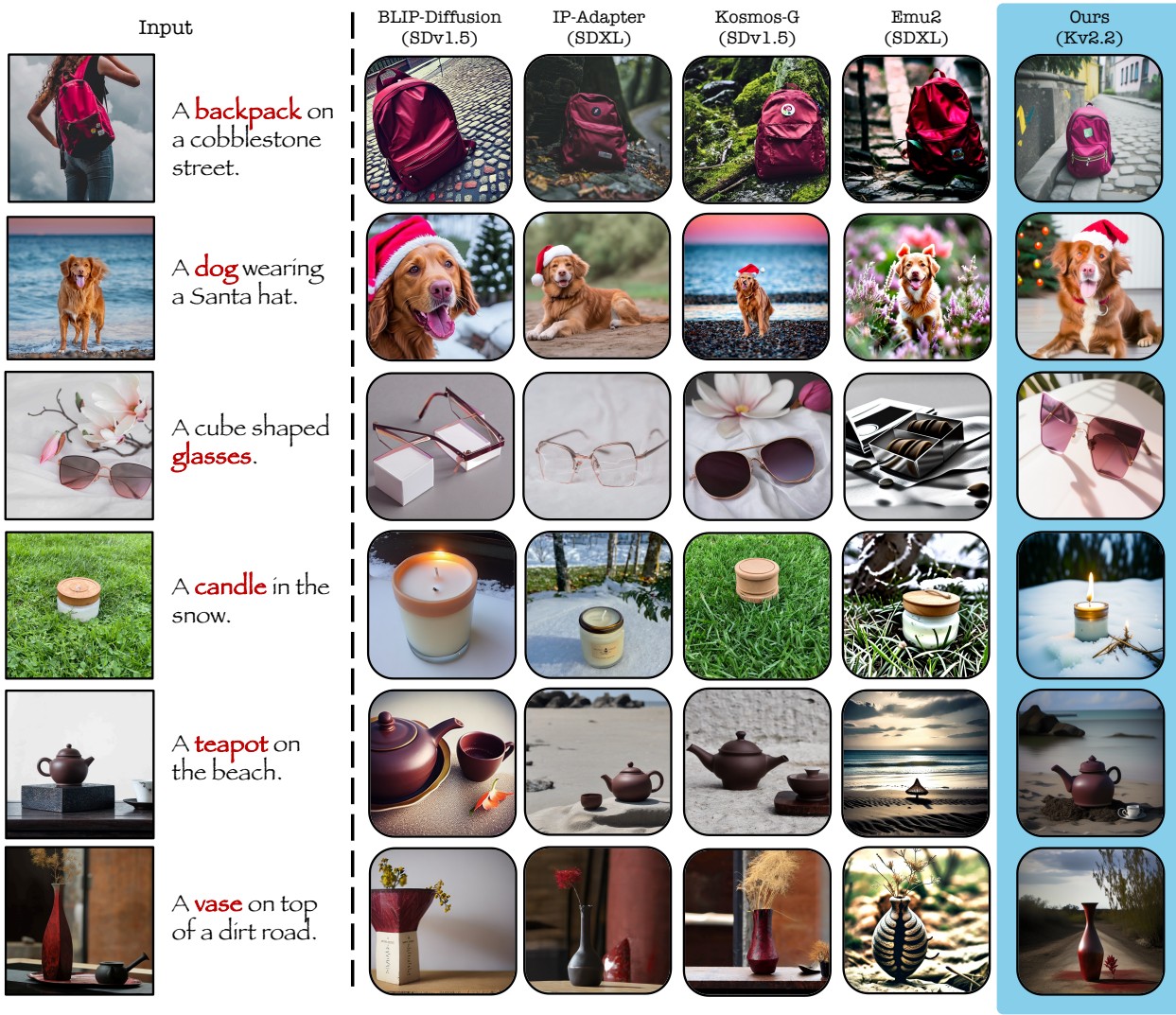

Figure 4: **This figure illustrates a qualitative comparison of $\lambda$-ECLIPSE with contemporary approaches for single-subject T2I generations, utilizing concepts and prompts from the Dreambench dataset.** For each method, concept, and prompt, we generate four images and select the one that most accurately represents the queried concept and composition.

### 3.3 Additional Concept-Specific Finetuning

Due to the nature of UnCLIP models, even if $\lambda$-ECLIPSE is very accurate, the diffusion UNet model ($h_\phi$) may not be effective in generating very unique visual representations. In Figure 3, we can observe that generated images do not always precisely follow the reference image (e.g. hair style of the person). However, such behavior is common across the fast P-T2I methods and they lack in terms of maintaining performance compared to the finetuning-based methods (as outlined in Table 2). Therefore, we extend the $\lambda$-ECLIPSE and perform concept-specific finetuning.

Compared to the traditional finetuning methodologies (e.g. DreamBooth), $\lambda$-ECLIPSE provides very unique advantages. As $\lambda$-ECLIPSE prior model ($f_\theta$) is pre-trained for personalization, there is no need for further finetuning the $f_\theta$ and we need to only finetune diffusion UNet model. Importantly, the fine-tuning of the $h_\phi$ does not depend on the text embeddings ($z_y$). Hence, this leads to stable fine-tuning of the $h_\phi$; unlike DreamBooth on stable diffusion that observes catastrophic forgetting. The new fine-tuning objective is:

Table 2: **Quantitative comparisons of different methodologies on Dreambench.** The **Bold** and underline represent the metric-specific first and second-ranked methods, respectively. * represents that we re-benchmark the performance from open-source weights.

| | Method | Base Model | Params | GPU Hours | DINO ($\uparrow$) | CLIP-I ($\uparrow$) | CLIP-T ($\uparrow$) |
|---|---|---|---|---|---|---|---|
| Finetuning | **Textual Inversion** | SDv1.5 | 768 | 1 | 0.569 | 0.780 | 0.255 |
| | **DreamBooth** | SDv1.5 | 0.9B | 0.2 | 0.668 | 0.803 | **0.305** |
| | **Custom Diffusion** | SDv1.5 | 57M | 0.2 | 0.643 | 0.790 | **0.305** |
| | **BLIP-Diffusion** | SDv1.5 | 0.9B | 0.1 | 0.670 | **0.805** | 0.302 |
| | $\lambda$-*ECLIPSE** | Kv2.2 | 0.9B | 0.2 | **0.682** | 0.796 | 0.304 |
| Finetuning-free | **Re-Imagen** | Imagen | - | - | 0.600 | 0.740 | 0.270 |
| | **ELITE** | SDv1.4 | 77M | 336 | 0.621 | 0.771 | **0.293** |
| | **Subject-Diffusion** | SDv1.5 | 252M | - | **0.711** | 0.787 | **0.293** |
| | **BLIP-Diffusion** | SDv1.5 | 1.5B | 2304 | 0.603 | 0.793 | 0.291 |
| | **IP-Adapter** | SDv1.5 | 22M | 672 | 0.629 | **0.827** | 0.264 |
| | **IP-Adapter** | SDXL | 22M | 672 | 0.613 | 0.810 | 0.292 |
| | **Kosmos-G** | SDv1.5 | 1.9B | 12300 | **0.618** | **0.822** | 0.250 |
| | **Emu2** | SDXL | 37B | - | 0.563 | 0.765 | 0.273 |
| | $\lambda$-*ECLIPSE** | Kv2.2 | **34M** | **74** | 0.613 | 0.783 | **0.307** |

$$\mathcal{L}_{decoder} = \mathop{\mathbb{E}}_{\substack{\epsilon \sim N(0,I) \\ t \sim [0,T], (z_x)}} \left[ ||\epsilon - h_\phi(x^{(t)}, t, z_x)||_2^2 \right]. \tag{2}$$

Here, $z_x$ is the visual feature of the reference concept image $x$. Notably, we do not need to use regularization from the DreamBooth as text alignment is already ensured during the pretraining stage of $\lambda$-*ECLIPSE*. Moreover, this finetuning can be performed across the set of given visual concepts altogether in a single model without degrading performance.

In summary, the prior model, trained with our image-text interleaved data and supplementary condition, presents an efficient pathway for resource-efficient multi-subject-driven image generations.

## 4 Experiments

In this section, we first introduce the experimental and evaluation setups. Later, we delve into the qualitative and quantitative results.

**Training and inference details.** We initialize our model, $\lambda$-*ECLIPSE*, equipped with 34M parameters. We train our model on an image-text interleaved dataset of 2M instances, partitioned into 1.6M for training and 0.4M for validation. The model is specifically tuned for the Kandinsky v2.2 diffusion image decoder. Therefore, we use pre-trained OpenCLIP-ViT-G/14[2] as the text and vision encoders, ensuring alignment with Kandinsky v2.2 image embeddings. Training is executed on 2 x A100 GPUs, leveraging a per-GPU batch size of 512 and a peak learning rate of 0.00005, across approximately 100,000 iterations, summing up to 74 GPU hours. During inference, the model employs 50 DDIM steps and 7.5 classifier-free guidance for the Kandinsky v2.2 diffusion image generator. Adhering to baseline methodologies, we perform the P-T2I following the baseline papers' protocols. For $\lambda$-*ECLIPSE*, target subject pixel regions in reference images are segmented before embedding extraction via the CLIP(vision) encoder. We drop the Canny edge map during inference unless specified explicitly. Unless specified all results (quantitative and qualitative) are without concept-specific additional fine-tuning.

**Evaluation setup.** We primarily utilize Dreambench (encompassing 30 unique concepts with 25 prompts per concept) for qualitative and quantitative evaluations using DINO and CLIP-based metrics Ruiz et al.

---

[2]https://huggingface.co/laion/CLIP-ViT-g-14-laion2B-s12B-b42K

Table 3: **Quantitative comparisons of different methodologies on ConceptBed.** We present results on $CCD$ ($\downarrow$) across three evaluation categories. The **Bold** and underline represent the metric-specific first and second-ranked methods, respectively. * represents our benchmarking.

| Method | Base Model | Concept Replication ($\downarrow$) | Concept Alignment ($\downarrow$) | Composition Alignment ($\downarrow$) | Average ($\downarrow$) |
|---|---|---|---|---|---|
| **Textual Inversion** | SDv1.4 | **0.0662** | **0.1163** | 0.1436 | **0.1087** |
| **Dreambooth** | SDv1.4 | 0.0880 | 0.3551 | 0.0360 | 0.1597 |
| **Custom Diffusion** | SDv1.4 | 0.2309 | 0.4882 | **0.0204** | 0.2465 |
| **ELITE*** | SDv1.4 | 0.3195 | 0.4666 | 0.1832 | 0.3231 |
| **BLIP-DIffusion*** | SDv1.5 | 0.3510 | **0.3245** | 0.1589 | 0.2781 |
| **IP-Adapter*** | SDXL | 0.3665 | 0.3571 | 0.0641 | 0.2626 |
| *λ-ECLIPSE** | Kv2.2 | **0.2853** | 0.3619 | **-0.0200** | **0.2091** |

Table 4: **Quantitative comparisons of different methodologies on Multibench.** The **Bold** and underline represent the metric-specific first and second-ranked methods on each metric, respectively.

| | Two Subjects | | | Three Subjects | |
|---|---|---|---|---|---|
| | Kosmos-G | Emu2 | *λ-ECLIPSE* | Emu2 | *λ-ECLIPSE* |
| **DINO** ($\uparrow$) | **0.4549** | 0.4451 | 0.4478 | 0.3168 | **0.3420** |
| **CLIP-I** ($\uparrow$) | **0.7759** | 0.7397 | 0.7409 | 0.6231 | **0.6463** |
| **CLIP-T** ($\uparrow$) | 0.2493 | 0.2673 | **0.3327** | 0.2819 | **0.3469** |

Table 5: **Quantitative results of Canny edge controlled P-T2I of different methodologies on Dreambench.** The **Bold** and underline represent the metric-specific first and second-ranked methods, respectively. Red highlighted numbers indicate the relative percentage drop for concept alignment compared to Table 2.

| Method | DINO ($\uparrow$) | CLIP-I ($\uparrow$) | CLIP-T ($\uparrow$) |
|---|---|---|---|
| **BLIP-Diffusion*** | 0.4234<sub style="color:red">29.7% | 0.7119 | 0.3152 |
| **IP-Adapter*** | 0.4281<sub style="color:red">31.9% | 0.7315 | 0.3034 |
| *λ-ECLIPSE** | **0.5173**<sub style="color:red">14.3% | **0.7437** | **0.3158** |

(2023a). Due to their limitations, we extend our evaluations on the ConceptBed Patel et al. (2023a) benchmark (covering 80 diverse imagenet concepts and a total of 33K composite prompts), where we report performance on concept replication, concept, and composition alignment using the Concept Confidence Deviation ($CCD$) metric Patel et al. (2023a). We extend Dreambench for multi-subject customization and present the Multibench dataset. Multibench contains about 24 unique concepts and 15 diverse prompts that result in 904 two-subject specific prompts and 1476 three-subject specific prompts. We provide further details about the Multibench in the appendix.

## 4.1 Results & Analysis

**Quantitative comparison.** The quantitative assessments detailed in Table 2 and Table 3 focus on the single-concept T2I task, while Table 4 shows the results on multi-concept-driven image generation. For Dreambench and Multibench, we generate and evaluate four images per prompt, reporting average performance on three metrics (DINO, CLIP-I, and CLIP-T). In the case of ConceptBed, we process each of the 33K prompts to generate a single concept image. The results, as depicted in these tables, highlight *λ-ECLIPSE*'s superior performance in composition alignment while maintaining competitive concept alignment. Analysis on ConceptBed (Table 3) indicates that *λ-ECLIPSE* exhibits a notable proficiency in concept replication, albeit with a marginal trade-off in concept alignment for enhanced composition fidelity. Com-

paratively, all baselines prioritize concept alignment, often at the expense of composition alignment. While $\lambda$-*ECLIPSE* improves the CLIP-T while preserving the DINO; achieved with significantly fewer resources. Notably, Multibench results (Table 4) indicate that $\lambda$-*ECLIPSE* significantly outperforms the Kosmos-G (2B params) and Emu2 (37B params) in terms of CLIP-T while maintaining the DINO performance. Therefore, we can conclude that $\lambda$-*ECLIPSE* is the most resource-efficient compared, especially when compared to the MLLM-based methods.

**Auxiliary finetuning.** We further perform concept-specific fine-tuning (as described in Section 3.3 and Section D). After finetuning, as shown in Table 2, $\lambda$-*ECLIPSE* outperforms the DreamBooth and BLIP-Diffusion in terms of concept alignment (DINO) while maintaining the performance on composition alignment (CLIP-T). Our findings, illustrated in Figure 5, reveal that $\lambda$-*ECLIPSE* and DreamBooth exhibit improved performance with incremental fine-tuning steps. Notably, the DINO score improved from 0.61 to 0.68 with few optimization steps and outperforms the baselines (see Table 2). A detailed analysis indicates that while DreamBooth's DINO score improves, its CLIP-T performance diminishes, hinting at concept overfitting. Conversely, $\lambda$-*ECLIPSE* consistently improves in DINO scoring without adversely impacting the CLIP-T performance, underscoring the efficacy of our image-text interleaved training approach at the prior stage. Qualitative comparisons, as shown in Figure 10, further highlight the benefits of fine-tuning $\lambda$-*ECLIPSE* with minimal steps. We provide detailed experimental setup in Appendix Section D.

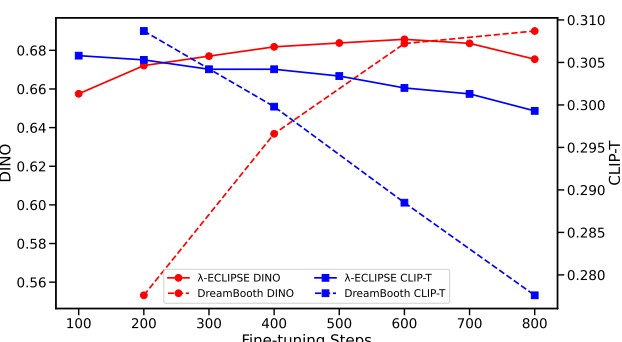

Figure 5: **DreamBooth (Stable Diffusion v1.5) *vs.* $\lambda$-*ECLIPSE* (with fine-tuning)** *w.r.t.* DINO and CLIP-T metrics on Dreambench.

**Qualitative comparisons.** In Figure 4, we present a range of single subject-specific images generated by various methodologies including BLIP-Diffusion, IP-Adapter, Kosmos-G, Emu2, and $\lambda$-*ECLIPSE*. $\lambda$-*ECLIPSE* demonstrates exemplary proficiency in composition while ensuring concept alignment. In contrast, the baselines often overemphasize reference images or exhibit concept dilution, leading to higher concept alignment but compromised composition. Interestingly, we find that Emu2 can capture the single-subjects but it fails to reproduce them with complex text compositions (as shown in Figure 4). Similarly, Figure 6a exhibits $\lambda$-*ECLIPSE*'s multi-concept generation prowess, in comparison to ZipLoRA (fine-tuning-based approach) along with Kosmos-G and Emu2 (Multimodal LLM-based approaches), underscoring its capability to rival compute-intensive methods. We discuss additional examples and limitations in the appendix. That said, even though $\lambda$-*ECLIPSE* improves the performance over the baselines, this is still not enough and it signifies the challenges associated with fast multi-concept personalization.

**Canny edge controlled image generation.** As shown in Figure 6b, the baseline (BLIP-Diffusion) adheres strictly to the imposed edge maps, often at the cost of concept retention (rows 1, 3, and 4). This leads to a large number of unwanted artifacts in the generated images. To further ground this behavior, we first generated images using Stable Diffusion v1.5 for Dreambench prompts without customization then we extracted the Canny edge map and used this edge map to control the subject-driven image generations. At last, we report the performance in Table 5. It can be observed that both baselines IP-Adapter and BLIP-Diffusion drop the DINO score by 30%, which follows the qualitative results. While $\lambda$-*ECLIPSE* do not follow the Canny edge precisely but preserves the concept alignment and improves the performance relatively by 21%.

**Ablations.** We extend our study to evaluate the individual contributions of different components in $\lambda$-*ECLIPSE*. Initially, the model's performance with solely the projection loss (referenced in Eq.1) is assessed. Subsequent experiments involve training $\lambda$-*ECLIPSE* variants with varying hyperparameters for the contrastive loss, specifically $\lambda$ values of 0.2 and 0.5. A comparative analysis of these baselines is conducted

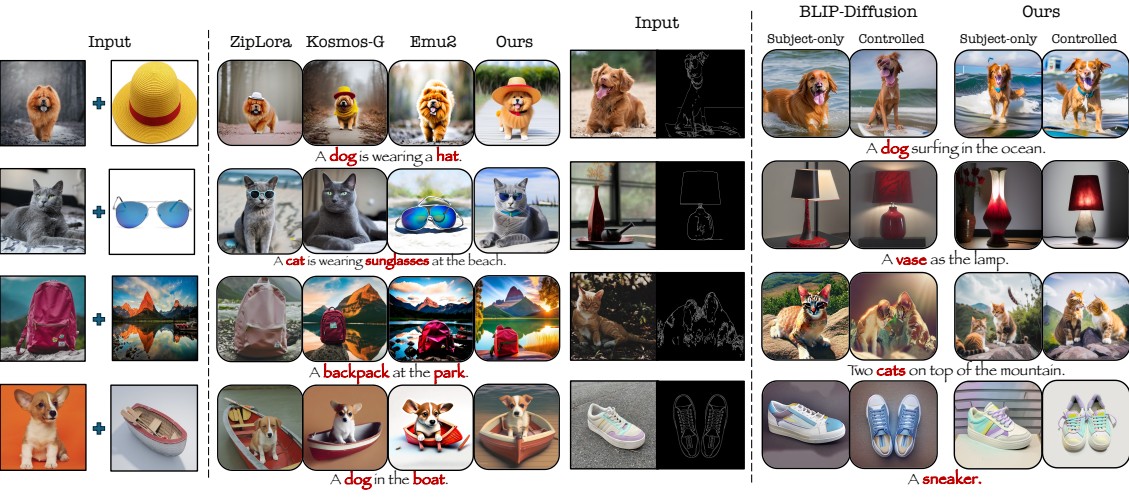

(a) **Multi-subject P-T2I**.                    (b) **Edge-guided P-T2I**.

Figure 6: **Qualitative comparison** between $\lambda$-*ECLIPSE* and other baselines.

Table 6: **Ablation studies** w.r.t. to the key components of $\lambda$-*ECLIPSE* design. We report the concept and composition alignment for single-subject T2I using $CCD$ ($\downarrow$) on the ConceptBed benchmark.

| Method | Concept Alignment ($\downarrow$) | Composition Alignment ($\downarrow$) |
|---|---|---|
| Projection loss (i.e. $\lambda$=0.0) | 0.394 | 0.008 |
| w/ contrastive loss ($\lambda$=0.5) | 0.435 | **-0.043** |
| w/ contrastive loss ($\lambda$=0.2) | 0.402 | -0.026 |
| w/ edge conditions ($\lambda$=0.2) | **0.362** | -0.020 |

against the fully equipped $\lambda$-*ECLIPSE* model, which incorporates $\mathcal{L}_{prior}$ (Eq.1) with $\lambda = 0.2$ and utilizes Canny edge maps during training. Relying solely on projection loss results in high concept alignment but compromises compositions (Table 6). The contrastive loss variant with $\lambda = 0.5$ enhances composition alignment at the expense of concept alignment, whereas $\lambda = 0.2$ achieves a more balanced performance. Crucially, the integration of Canny edge maps during training optimally balances both alignments and, specifically, improves the concept alignment. The negative values indicate that the $CCD$ oracle model is highly confident in the generated images.

**Multi-subject interpolation.** A key attribute of the CLIP latent space is the ability to perform smooth interpolation between two sets of embeddings. We conducted experiments to demonstrate $\lambda$-*ECLIPSE*'s ability to learn and replicate this smooth latent space transition. We selected two distinct dog breeds (<dog1>, <dog2>) and two types of hats (<hat1>, <hat2>) as the concepts. $\lambda$-*ECLIPSE* was then used to estimate image embeddings for all four possible combinations, each corresponding to the input phrase "a <dog> wearing a <hat>." Fig. 7 showcases a gradual and seamless transition in the synthesized images from the top left to the bottom right. Unlike current diffusion models, which often exhibit sensitivity to input variations requiring numerous iterations of user interactions for desired outcomes, $\lambda$-*ECLIPSE* inherits CLIP's smooth latent space. This not only facilitates progressive changes in the conceptual domain but also extends the model's utility by enabling personalized **multi-subject interpolations**.

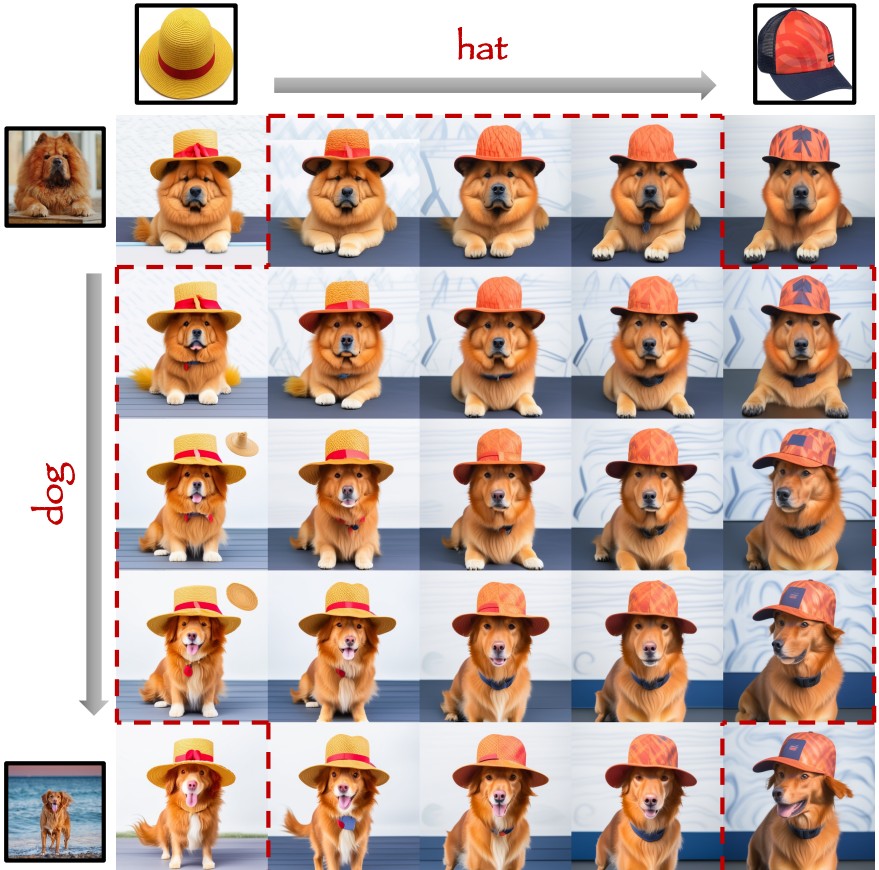

Figure 7: **Interpolation between four concepts.** Here, we estimate the image embedding using $\lambda$-*ECLIPSE* corresponding to each corner and then interpolate from top to bottom and left to right. At last, we use the Kandinsky v2.2 diffusion UNet model to generate the images with fixed random seeds from these sets of image embeddings.

## 5    Conclusion

In this paper, we have introduced a novel training-time diffusion-agnostic methodology for personalized text-to-image (P-T2I) applications, utilizing the latent space of the pre-trained CLIP model with high efficiency. Our $\lambda$-*ECLIPSE* model, trained on an image-text interleaved dataset, achieves the capability to execute single-concept, multi-concept, and edge-guided controlled P-T2I tasks using a singular model framework, while simultaneously minimizing resource utilization. Notably, $\lambda$-*ECLIPSE* sets a new benchmark in achieving competitive performance in terms of concept and composition alignment. Furthermore, our research illuminates the potential of $\lambda$-*ECLIPSE* in exploring and leveraging the smooth latent space. This capability unlocks new avenues for interpolating between multiple concepts, thereby generating entirely novel concepts. Our findings underscore a promising pathway to improve MLLMs to effectively control the pre-trained diffusion image models without necessitating extra supervision.

### Limitations

Primarily, despite its strengths, CLIP's inability to perfectly capture hierarchical representations adds the upper bound on performance. Hence, enhancing CLIP's representations could significantly boost our framework's efficacy in P-T2I mapping. Even though $\lambda$-*ECLIPSE* model, trained on 34 million parameters and 1.6 million images, presents a substantial foundation, yet, there's potential for further refinement, as increasing the quality of data and the number of parameters could yield even better outcomes. However, this is outside

the scope of this paper. Additionally, we validate the $\lambda$-`ECLIPSE` on only upto three concepts (due to the real-life usecases) and adding more concepts could be explored in future works.

## Broader Impact

The current landscape of text-to-image (T2I) generative models is dominated by approaches that rely on extensive data and large-scale models to achieve state-of-the-art (SOTA) performance, which demands significant computational resources. In contrast, our work with $\lambda$-`ECLIPSE` demonstrates that it is feasible to attain competitive performance relative to SOTA large models while achieving a tenfold reduction in resource consumption. This advancement not only makes T2I generative models more accessible and cost-effective but also promotes sustainable AI practices by significantly lowering the environmental impact associated with large-scale model training.

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

# A    Preliminaries for T2I Diffusion Models

As evidenced in numerous contemporary studies regarding T2I models, Stable Diffusion (SD) Rombach et al. (2022) has emerged as a predominant backbone for T2I models. SD involves training diffusion models in latent space, reversing a forward diffusion process that introduces noise into the image. A notable feature of SD is its integration of cross-attention, facilitating various conditions like text input. Operating in VQ-VAE latent space, SD not only achieves exceptional generative performance surpassing that in pixel space but also significantly reduces the computational demands.

UnCLIP models (such as DALL-E 2) are very similar to the Stable Diffusion. In contrast, the UnCLIP takes the modular approach. UnCLIP first trains the diffusion text-to-image to the image prior ($f_\theta$) to estimate the image embeddings ($z_x$) from the text embeddings ($z_y$). In parallel, a UNet-like diffusion image generator ($h_\phi$) is trained to generate images ($x$) conditioned on ground truth vision embeddings ($z_x$).

Traditionally, T2I prior is modeled to estimate $x_0$-prediction instead of $\epsilon$-prediction. Given the forward function $z_x^{(t)} \sim q(t, z_x)$, the goal of $f_\theta$ is to directly estimate $z_x$ for all timesteps $t \sim [0, T]$ as:

$$\mathcal{L}_{prior} = \mathop{\mathbb{E}}_{\substack{t \sim [0,T], \\ z_x^{(t)} \sim q(t, z_x)}} \left[ ||z_x - f_\theta(z_x^{(t)}, t, z_y)||_2^2 \right]. \tag{3}$$

*ECLIPSE* proposes the contrastive learning strategy (Eq. 1 – main paper) instead of minimizing Eq. 3. The diffusion image generator is trained by following standard $\epsilon$-prediction formulation. Here, $h_\phi$ will estimate the ground truth added Gaussian noise $\epsilon \sim N(0, I)$, given the noise image $X^{(t)}$ for all timesteps $t \sim [0, T]$ and input conditions (such as $z_x, z_y$).

$$\mathcal{L}_{decoder} = \mathop{\mathbb{E}}_{\substack{\epsilon \sim N(0,I) \\ t \sim [0,T], \\ (z_x,\ z_y)}} \left[ ||\epsilon - h_\phi(x^{(t)}, t, z_x, z_y)||_2^2 \right]. \tag{4}$$

For models like Kandinsky v2.2, we drop the $z_y$ to condition the model on $z_x$. Therefore, $\lambda$-*ECLIPSE* also only conditions the image generation with $z_y$ in the prior stage.

# B    Image-Text Interleaved Training Details

**Dataset Creation**    In constructing the dataset, we adhered to the data processing pipeline of Subject Diffusion Ma et al. (2023a). We utilized the LAION-5B High-Res dataset, requiring a minimum image size of 1024x1024 resolution. Original captions were replaced with new captions generated by BLIP-2 (flan-t5-xl)[3]. Subjects were extracted using Spacy[4]. For each subject, we identified bounding boxes employing Grounding DINO Liu et al. (2023a), setting both box-threshold and text-threshold values to 0.2. We retained images with 1 to 8 detected bounding boxes, discarding the rest. Additionally, captions with multiple instances of identical objects were filtered, allowing a maximum of 6 identical objects. Following bounding box detection, individual subject masks were isolated using Segment-Anything (SAM) Kirillov et al. (2023). To enhance the efficiency of the training process, we pre-processed the dataset by pre-extracting features from CLIP vision and text encoders. During this phase, images predominantly featuring a background (white portion) exceeding 50% of the total area were excluded. We preserved bounding boxes with an area ranging from 0.08 to 0.7 of the total image area and logit scores of at least 0.3. Masks constituting less than 40% of the bounding box area were discarded. For the selection of subjects in images, we constrained the range to 1-4 subjects per image, excluding those with more than 4 subjects. At last, the interleaved image-text examples with respective ground truth images are shown in Figure 9.

**Dataset Statistics**    In the final analysis, our dataset comprised a total of 1,990,123 images. The distribution of subjects per image exhibited a range from 1 to 4, with the following breakdown: 1,479,785 images

---

[3]https://huggingface.co/Salesforce/blip2-flan-t5-xl
[4]https://spacy.io

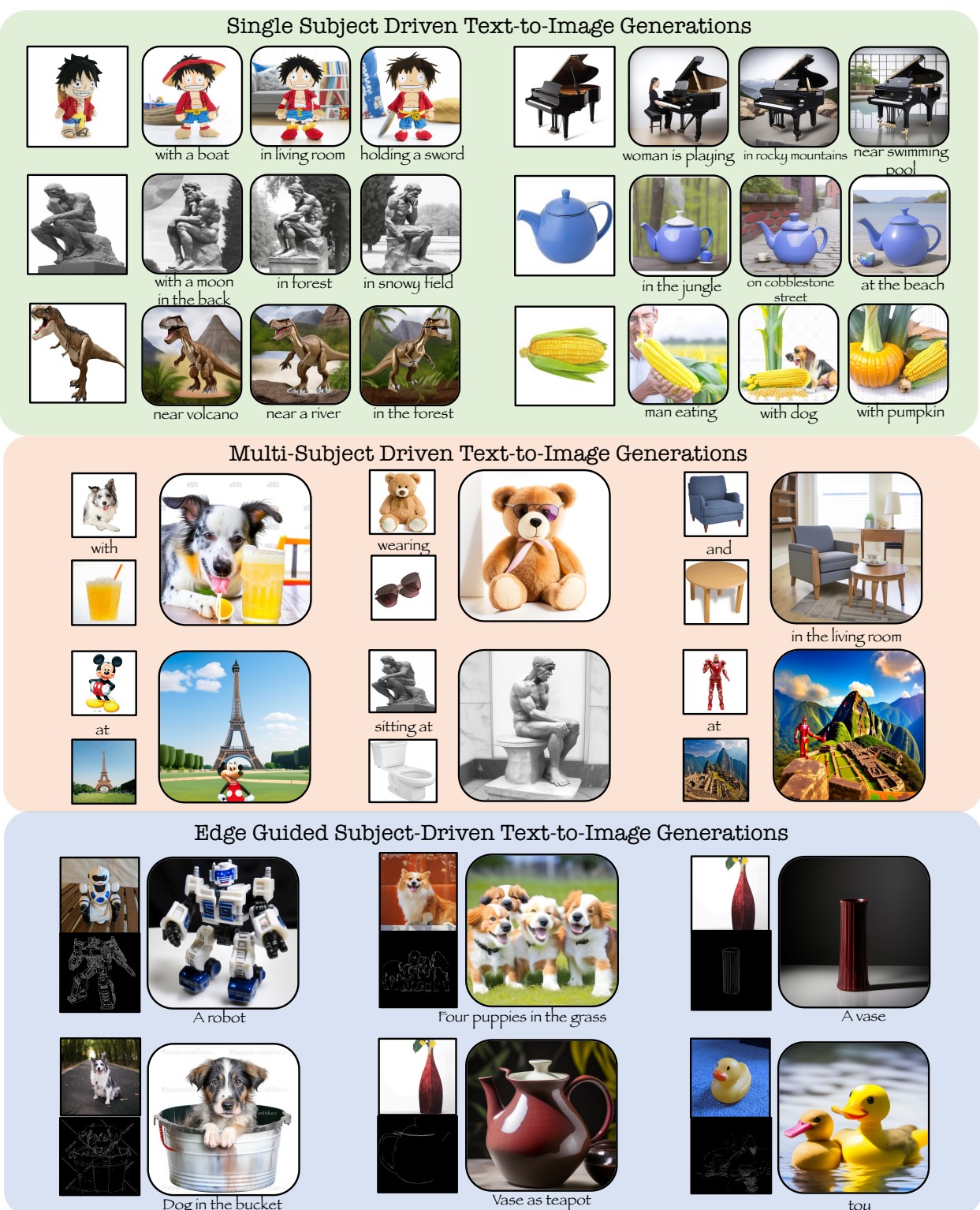

Figure 8: Qualitative results categorized by generative capabilities.

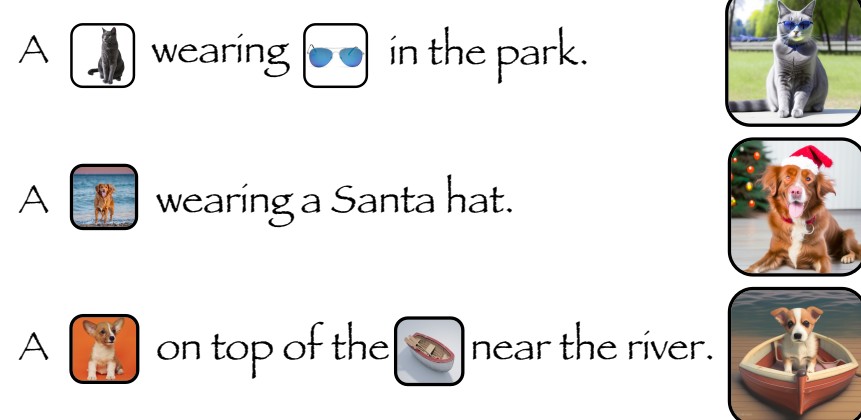

Figure 9: **Examples of image-text interleaved training data.** The left column shows the input of the prior model and the right images shows the ground truth corresponding images. Note: these examples are generated from $\lambda$-`ECLIPSE` for better interpretability.

featuring one subject, 432,831 images with two subjects, 65,597 images containing three subjects, and 11,910 images showcasing four subjects. The overall count of unique subjects acquired from this dataset amounted to 30,358. We partitioned our dataset into an 80:20 split between training and validation, reserving the remaining 1.6 million images for training and the rest for validation.

## C  Implementation Details

The $\lambda$-`ECLIPSE` transformer prior architecture is significantly more compact compared to other Text-to-Image (T2I) methodologies. Our model employs a configuration of 16 Attention Heads, each with a dimension size of 32, alongside a total of 10 layers. Additionally, the embedding dimension size for our model is set at 1280, supplemented by 4 auxiliary embeddings (including, one for canny edge map). As $\lambda$-`ECLIPSE` is not a diffusion prior model, we do not keep time embedding layers. Overall, the $\lambda$-`ECLIPSE` model comprises approximately 34 million parameters, establishing it as a streamlined yet effective solution for Personalaized-T2I. Notably, the standard UnCLIP T2I priors contain 1 billion parameters.

## D  $\lambda$-`ECLIPSE` with Finetuning

As demonstrated in the main paper (Table 2), the superiority of fine-tuning-based personalization methodologies, whether applied to single-subject or multi-subject frameworks, over non-fine-tuning alternatives is evident. Consequently, we have expanded our analysis through additional fine-tuning of the $\lambda$-`ECLIPSE`.

**Experimental Setup.**   Given that $\lambda$-`ECLIPSE` effectively trains the T2I prior, capturing concept-specific features to a significant degree, we opted not to further optimize this component. Our focus shifted to exclusively fine-tuning the diffusion UNet model ($h_\phi$), employing the AdamW optimizer at a learning rate of 1e-5, without warm-up steps. For the DreamBooth application within the Stable Diffusion v1.5 model, we selected a learning rate of 5e-6, maintaining consistency in other hyperparameters. To simplify, we excluded the use of a prior preservation regularizer and conducted training on the Dreambench platform using a single RTX A6000 GPU.

**Advantages of fine-tuning $\lambda$-`ECLIPSE`.**   The fine-tuning of $\lambda$-`ECLIPSE`, in comparison to the baselines, reveals two key benefits: 1) Achieving state-of-the-art (SOTA) performance within a few finetuning steps. 2) Unlike the Stable Diffusion model, which exhibits catastrophic forgetting of nearby concepts post-DreamBooth fine-tuning, $\lambda$-`ECLIPSE` maintains previous knowledge. This suggests that a single model is sufficient to effectively fine-tune across multiple concepts together.

Table 7: **The detailed overview of subject-driven text-to-image generative methodologies.** * represents the backbone base models listed are subject to potential updates or modifications.

| Method | Multi-Subject | Finetuning-Free | Base-Model | # of Input Images |
|---|---|---|---|---|
| Re-Imagen Chen et al. (2022) | ✗ | ✓ | Imagen | Single |
| Textual Inversion Gal et al. (2022) | ✗ | ✗ | SDv1.4 | Multiple |
| DreamBooth Ruiz et al. (2023a) | ✗ | ✗ | SDv1.4 | Multiple |
| Custom Diffusion Kumari et al. (2023) | ✓ | ✗ | SDv1.4 | Multiple |
| ELITE Wei et al. (2023) | ✗ | ✓ | SDv1.4 | Single |
| E4T Gal et al. (2023) | ✗ | ✗ | SD | Single |
| Cones Liu et al. (2023b) | ✓ | ✗ | SDv1.4 | Single |
| SVDiff Han et al. (2023) | ✓ | ✗ | SD | Multiple |
| UMM-Diffusion Ma et al. (2023b) | ✗ | ✓ | SDv1.5 | Single |
| XTI Voynov et al. (2023) | ✗ | ✗ | SDv1.4 | Multiple |
| Continual Diffusion Smith et al. (2023) | ✓ | ✗ | - | Multiple |
| InstantBooth Shi et al. (2023) | ✗ | ✓ | SDv1.4 | Multiple |
| SuTi Chen et al. (2023c) | ✗ | ✓ | Imagen | Multiple |
| Taming Jia et al. (2023) | ✗ | ✓ | Imagen | Single |
| BLIP-Diffusion Li et al. (2023a) | ✗ | ✓ | SDv1.5 | Single |
| Cones 2 Liu et al. (2023c) | ✓ | ✗ | SDv2.1 | Single |
| DisenBooth Chen et al. (2023a) | ✗ | ✗ | SDv2.1 | Single |
| FastComposer Xiao et al. (2023) | ✓ | ✓ | SDv1.5 | Single |
| Perfusion Tewel et al. (2023) | ✓ | ✗ | SDv1.5 | Multiple |
| Mix-of-Show Gu et al. (2023) | ✓ | ✗ | Chilloutmix | Multiple |
| NeTI Alaluf et al. (2023) | ✗ | ✗ | SDv1.4 | Mulitple |
| Break-A-Scene Avrahami et al. (2023) | ✓ | ✗ | SDv2.1 | Single* |
| ViCo Tumanyan et al. (2023) | ✗ | ✗ | SDv1.4 | Mulitple |
| Domain-Agnostic Arar et al. (2023) | ✗ | ✗ | - | Single |
| Subject-Diffusion Ma et al. (2023a) | ✓ | ✓ | SDv2 | Single |
| HyperDreamBooth Ruiz et al. (2023b) | ✗ | ✗ | SDv1.5 | Single |
| IP-Adapter Ye et al. (2023) | ✗ | ✓ | SDv1.5 | Single |
| Kosmos-G Pan et al. (2023) | ✓ | ✓ | SDv1.5 | Single |
| Zip-LoRA Shah et al. (2023) | ✓ | ✗ | SDXL | Multiple |
| CatVersion Zhao et al. (2023) | ✗ | ✗ | SDv1.5 | Multiple |
| SSR-Encoder Zhang et al. (2023b) | ✓ | ✓ | SDv1.5 | Single |
| Emu2 Sun et al. (2023) | ✓ | ✓ | SDXL | Single |
| $\lambda$-`ECLIPSE` (ours) | ✓ | ✓ | Kv2.2 | Single |

This analysis underscores the strategic advantages and enhanced efficiency of fine-tuning $\lambda$-`ECLIPSE` for personalized applications in complex visual data processing.

## E    Extended P-T2I Baselines Comparison

We further expand our comparative analysis of P-T2I methods encompassing a total of 33 approaches including ours and parallel works. Table 7 summarizes them into four crucial aspects: 1) multi-subject support, 2) fine-tuning free, 3) base model types, and 4) the required number of input images. To summarize, $\lambda$-`ECLIPSE` is the only methodology built on top of the UnCLIP models while supporting multi-subject driven image generation with fine-tuning free, and only requires a single reference image input for the training. We detail the comparison below:

**Multi-Subject Generation.**    Multi-subject generation enables users to integrate multiple personal subjects to generate an image that follows the text prompts and aligns with all the concept visuals. In total, 15 of the 33 methods offer this capability, while 6 methods support fast multi-subject personalization, others demand separate training for each subject to be learned and then an additional fusing step for combining the learned subjects is required (i.e. Zip-LoRA, Mix-of-Show). Among these methods, only a few can learn auxiliary guided information such as canny edge, depth maps, or open-pose and adapt style variation (i.e. Kosmos-G).

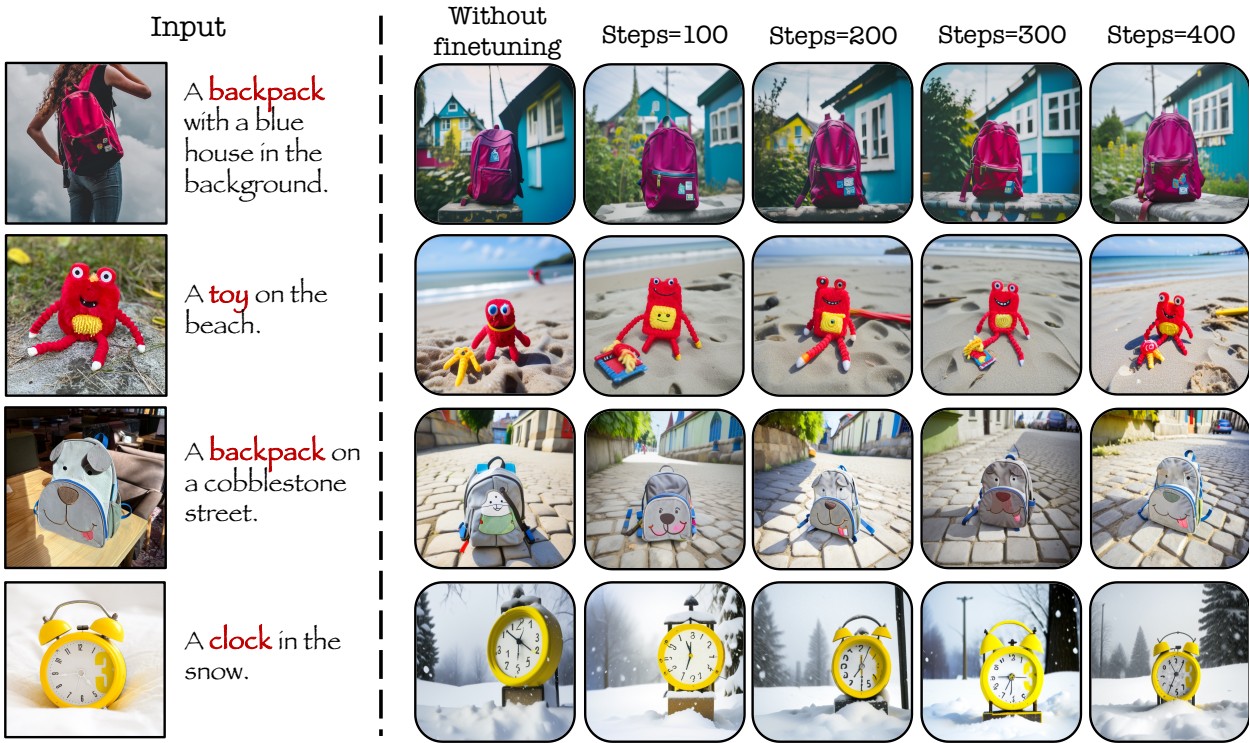

Figure 10: Qualitative examples of $\lambda$-`ECLIPSE` without finetuning and different stages of finetuning.

**Fine-tuning Free (Fast Personalization).**   Many methods require test-time fine-tuning. Each varies on which part alteration occurs, as early models tend to modify the whole UNet. In contrast, recent models tune a small portion of the cross-attention layers or introduce additional layers performing as adapters. In our analysis of P-T2I methodologies, 14 out of 33 methods employ a finetuning-free approach which enables fast personalization.

**Diffusion Independent.**   A majority of the reviewed models utilize diffusion models, with Stable Diffusion being the predominant choice, spanning versions 1.4, 1.5, 2.1, and XL. Few adapt Imagen (SuTi, Taming) and Mix-of-show employs ChillOutMix as their pre-trained model, known for its adeptness at preserving realistic concepts like human faces. A unique outlier in this landscape is our $\lambda$-`ECLIPSE`, the only one that eschews the use of any diffusion prior model.

**Easiness of Use.**   A more user-friendly model typically requires a single reference image per subject, as opposed to multiple images of the same subject. In our study, 19 methods offer P-T2I capabilities with just one input image. In contrast, others often require 4 to 5 images of the subject. Additionally, some methods necessitate storage space for learned concepts, ranging from a few hundred kilobytes (e.g., Perfusion, HyperDreamBooth) to several megabytes (e.g., Zip-LoRA). Our method stands out by eliminating the need for individual concept pre-learning or storing any artifacts for P-T2I utilization, offering a streamlined, efficient user experience.

## F   Multibench Dataset

We provide additional qualitative results in Figure 8. For the multi-subject image benchmark, our dataset comprises 2,308 unique prompts, segmented into 904 two-subject and 1,476 three-subject prompts. This dataset integrates subjects from the original DreamBench dataset, featuring 30 distinct concepts. We expanded the dataset by incorporating additional concepts vital for two and three subject-specific prompts, such as various parks, hats, glasses, and more. Prompt templates and the count of unique subject cat-

Table 8: **Example of prompt templates used for Multibench dataset.** Subjects presented in Table 9 are placed in {}.

| Two subjects | Three subjects |
|---|---|
| {} in the {} | {} with a {} and {} |
| {} wearing a {} | {} is playing with {} in {} |
| {} chasing a {} | {} with {} in front of {} |
| {} looking at a {} | {} with a {} and a view of the {} |
| {} is sitting on a {} | {} with a {} and {} in the background |
| {} standing on a {} | |
| {} and {} playing in the garden | |
| {} and {} on top of the mountain | |
| {} and {} in the jungle | |
| {} and {} in the snow | |
| {} and {} on the beach | |
| {} and {} on a cobblestone street | |
| {} and {} standing next to each other | |

Table 9: **Number of occurrences of unique subject categories.** The left side of the table are subjects used for two subjects prompts, and the right side of the table are subjects used for three subjects prompts.

| Two subjects | | | | Three subjects | | | |
|---|---|---|---|---|---|---|---|
| dog | 76 | boat | 5 | dog | 81 | rainbow | 35 |
| cat | 76 | park | 4 | stuffed animal | 105 | ruins | 35 |
| bird | 76 | ruins | 9 | toy | 105 | tower | 35 |
| horse | 73 | castle | 5 | cat | 81 | horse | 81 |
| guinea pig | 73 | desert | 4 | desert | 60 | bird | 81 |
| glasses | 5 | rainbow | 5 | hill | 60 | guinea pig | 81 |
| hat | 5 | candle | 5 | castle | 45 | guitar | 25 |
| tower | 10 | backpack | 3 | backpack | 65 | french horn | 25 |
| | | | | can | 130 | vase | 25 |
| | | | | candle | 65 | robot | 25 |
| | | | | church | 35 | | |

egories featured in prompts are detailed in Tables 8 and 9, respectively. Overall, the dataset includes 217 two-subject compositions and 405 three-subject compositions, enriching the benchmark's diversity and comprehensiveness.

# G Additional Ablations

In this section, we perform extra ablations studies. Specifically, first, we extend the $\lambda$-*ECLIPSE* to other pretrained diffusion models (namely, Stable UnCLIP). Later, we analyze the impact of interleaved pertaining in terms of qualitative and quantitative evaluations. Finally, we perform an analysis by varying the data and model size.

## G.1 Generalization to Pretrained UnCLIP Diffusion Decoders

Our approach is designed to generalize across any pretrained UnCLIP diffusion models, including Stable-UnCLIP/SDv2.1, Karlo, and Kandinsky v2.2. We conducted additional pretraining experiments to substantiate this claim and demonstrated the generalization ability of $\lambda$-*ECLIPSE*. We would like to reiterate the core pipeline of $\lambda$-*ECLIPSE* or UnCLIP models, as detailed in Appendix A. The UnCLIP model comprises two key modules: (1) the Prior model, which maps the text embedding to the image embedding, and (2) the

Table 10: **Ablation study on generalization ability of $\lambda$-`ECLIPSE` with respect to different pretrained UnCLIP models.** We report performance on DreamBench dataset.

| Method | DINO (↑) | CLIP-I (↑) | CLIP-T (↑) |
|---|---|---|---|
| **Stable UnCLIP (Stable Diffusion v2.1)** | 0.564 | 0.778 | 0.276 |
| **Kandinsky v2.2** | **0.613** | **0.783** | **0.307** |

Table 11: **Ablation on the effect of interleaved data for training $\lambda$-`ECLIPSE`.** We report performance on DreamBench dataset.

| # of Subjects | Metric | w Interleaved | w/o Interleaved | Difference |
|---|---|---|---|---|
| 1 | DINO | **0.6130** | 0.6006 | **-2.02%** |
|  | CLIP-I | 0.7830 | **0.7882** | 0.66% |
|  | CLIP-T | **0.3070** | 0.3048 | **-0.72%** |
| 2 | DINO | **0.4478** | 0.4332 | **-3.26%** |
|  | CLIP-I | **0.7409** | 0.7384 | **-0.34%** |
|  | CLIP-T | **0.3327** | 0.3271 | **-1.68%** |
| 3 | DINO | **0.3420** | 0.3202 | **-6.37%** |
|  | CLIP-I | 0.6463 | **0.6480** | 0.26% |
|  | CLIP-T | 0.3469 | 0.3469 | 0.00% |

Diffusion rendering model, which synthesizes the image based on the estimated image embeddings from the prior model. For our generalization studies, we adhered to two essential criteria:

- Selection of an open-source UnCLIP (2 stage) model, where we chose Stable-UnCLIP, a modified version of Stable Diffusion v2.1, to accept image embeddings as input.

- Choice of pretrained CLIP model, with Stable-UnCLIP utilizing the OpenAI-CLIP-ViT-L/14 model. Accordingly, we trained $\lambda$-`ECLIPSE` with this model as a preliminary feature extractor.

Following these guidelines, we trained $\lambda$-`ECLIPSE` with the same parameters and evaluated its performance on DreamBench (see Table 10). Our results confirm that $\lambda$-`ECLIPSE` effectively generalizes to different diffusion models and CLIP variants. Notably, the Stable-UnCLIP variant of $\lambda$-`ECLIPSE` achieves performance similar to Emu2, reinforcing the superiority of our initial design choices.

### G.2 Impact of Interleaved Pretraining

Section 3.2 mentions that training $\lambda$-`ECLIPSE` without interleaved pretraining would yield similar results for "single-concept" P-T2I tasks. However, for "multi-concept" personalization, our experiments reveal that models trained without interleaved data sometimes struggle to synthesize the desired images. We conducted additional experiments by training the model without interleaved data. Specifically, we concatenated the prompt embedding from the CLIP text encoder with the concept-specific image embedding and trained the model with identical hyperparameters. The performance comparisons on DreamBench (single concept) and Multibench (multiple concepts) are shown below. Our findings indicate that without interleaved data, the model's ability to align concepts decreases as the number of concepts increases, resulting in a sharp 6% drop in the DINO score (see Table 11). This performance degradation is primarily due to attribute leakage. As shown in Figure 11, when the model is tasked with generating "A backpack at the ruins" + <backpack> + <ruins>, the non-interleaved pretrained models tend to generate the backpack with the color of the ruins.

Table 12: **Ablation study on pretraining data size.** We report DreamBench performance of $\lambda$-*ECLIPSE* models trained on 100k, 500k, 1M and 2M interleaved image-text pairs. $^*$ denotes the proposed $\lambda$-*ECLIPSE* model checkpoint.

| Data Size | DINO ($\uparrow$) | CLIP-I ($\uparrow$) | CLIP-T ($\uparrow$) |
|---|---|---|---|
| **100K** | 0.593 | 0.786 | 0.301 |
| **500K** | 0.595 | 0.777 | 0.305 |
| **1M** | 0.596 | 0.778 | 0.306 |
| **2M**$^*$ | **0.613** | **0.783** | **0.307** |

Table 13: **Ablation study on pretraining model size.** We report the DreamBench performance of $\lambda$-*ECLIPSE* models trained on 5M, 34M, and 70M parameters. $^*$ denotes the proposed $\lambda$-*ECLIPSE* model checkpoint.

| Model Size | DINO ($\uparrow$) | CLIP-I ($\uparrow$) | CLIP-T ($\uparrow$) |
|---|---|---|---|
| **5M** | 0.586 | 0.779 | 0.305 |
| **34M**$^*$ | **0.613** | **0.783** | 0.307 |
| **70M** | 0.593 | 0.775 | **0.309** |

### G.3 Effect of Data and Model Sizes

We also conduct ablation on data sizes (100k, 500k, 1M, and 2M) and model parameter sizes (5M, 35M, and 70M). Tables 12 and 13 report the performance of these newly trained models on DreamBench. All models were trained with identical hyperparameters as the proposed $\lambda$-*ECLIPSE*, though this may not fully optimize the larger models (e.g., 70M parameters). The results show that data size influences model performance, with larger datasets improving concept understanding and prompt compositions. This also follows the qualitative results in Figure 12. Notably, $\lambda$-*ECLIPSE* with only 5M parameters deliver performance close to that of larger models, and the 34M parameter model even surpasses the 70M parameter model in terms of DINO score. However, qualitative results (see Figure 13) show that increasing model parameters enhances qualitative performance and concept alignment. Specifically, the 70M parameter model excels in generating finer details of the reference concept while adhering closely to text prompts. The 34M model offers a more balanced trade-off between performance and resource efficiency.

## H  Qualitative Results & Failure Cases

In this section, we showcase a collection of detailed qualitative examples from the P-T2I generation process, highlighting the challenges of crafting complex compositions within $\lambda$-*ECLIPSE* and comparative models. As depicted in Figure 14, the complexity of the showcased examples progressively increases, illustrating a noticeable escalation in the intricacy of visual concepts from the top to the bottom of the figure. With the rising complexity, we note a universal decline in the ability of all methodologies, including $\lambda$-*ECLIPSE*, to preserve subject fidelity accurately. Interestingly, despite these challenges, $\lambda$-*ECLIPSE* demonstrates a better grasp of compositional integrity, unlike the baseline models which falter across all complexity levels.

Moreover, we present instances demonstrating the variability in outcomes produced by P-T2I methods across different trials. As illustrated in Figure 15, while there is a semblance of consistency in generating single and multiple concepts between models, Kosmos-G specifically shows variability in rendering multiple concepts—occasionally misplacing elements of the Ironman suit on a dog or failing to include it altogether. This phenomenon suggests that $\lambda$-*ECLIPSE* minimizes image diversity to enhance result consistency, a trait observed across the UnCLIP model family.

Figure 10 offers qualitative insights into the performance of $\lambda$-*ECLIPSE* without and with minimal fine-tuning. It is evident that in certain edge cases, where $\lambda$-*ECLIPSE* initially struggles to fully grasp novel visual concepts without finetuning, a modest application of few optimization iterations significantly enhances

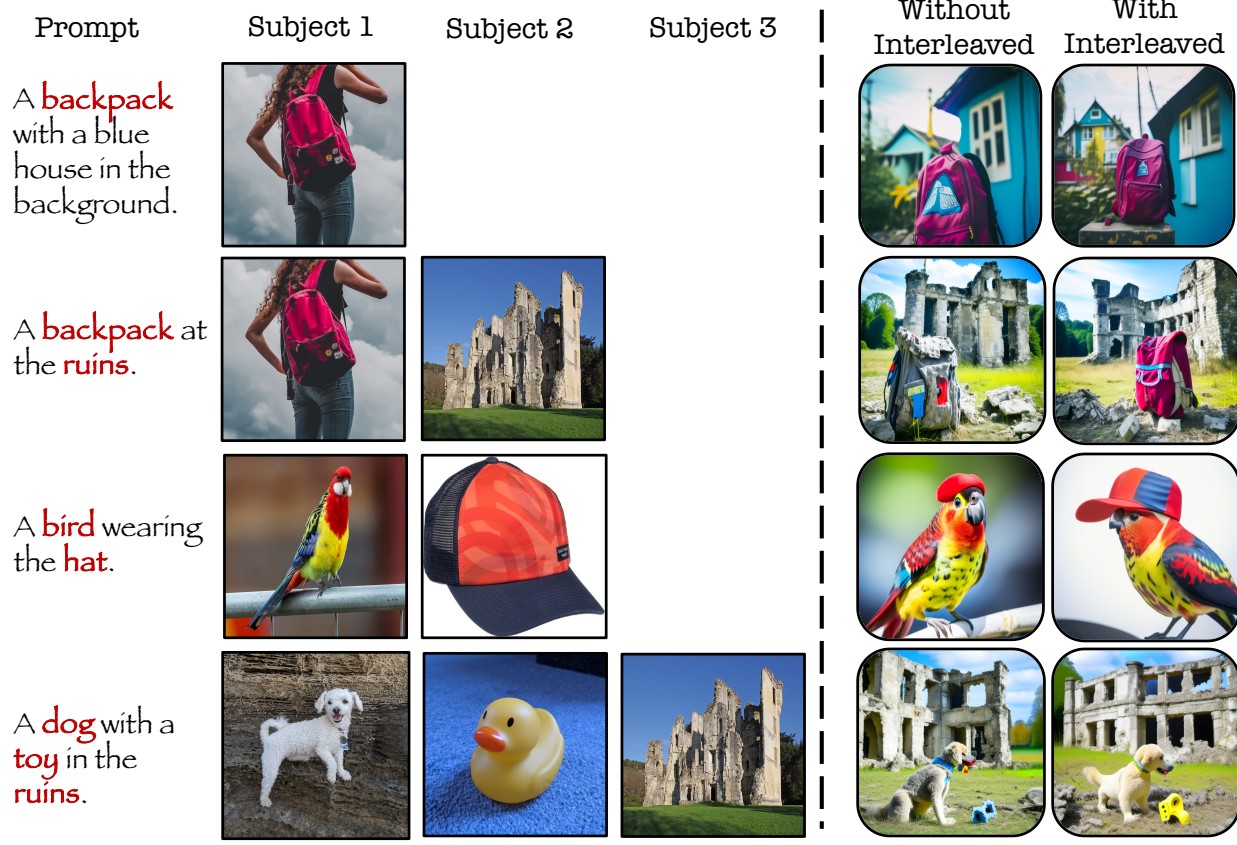

Figure 11: Qualitative examples $\lambda$-`ECLIPSE` model trained with and without the interleaved pretraining strategy.

concept capture. Further optimization not only preserves text composition but also enriches minor, subject-specific details, underscoring the adaptability and finesse of $\lambda$-`ECLIPSE` in nuanced image generation.

Moreover, in our evaluations using the Multibench dataset, we noticed that both the baseline models (Kosmos-G and Emu2) and $\lambda$-`ECLIPSE` encounter difficulties in precisely maintaining all subject-specific details, as depicted in Figure 17. **This underscores that zero-shot multi-subject P-T2I generation remains a significant challenge in the field.** Further, we explored how well each model preserves genuine human facial characteristics in various scenarios, particularly when combined with differing captions. The qualitative examples in Figure 16 shed light on this aspect. Although each model strives to maintain the original facial features, none succeeds in replicating the specific personal facial details accurately. These instances typically fall short of precisely conveying the intended compositions, with the exception of one scenario in IP-Adapter FaceID, indicating a notable area for future improvements in model performance.

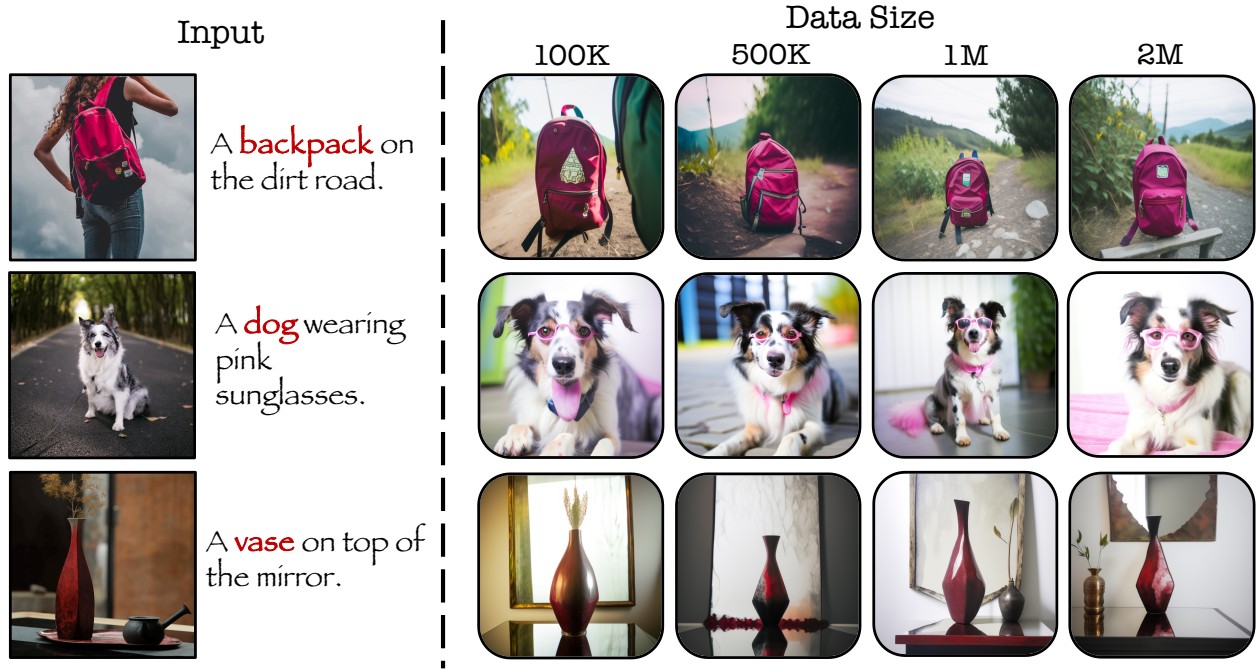

Figure 12: Qualitative examples $\lambda$-*ECLIPSE* model trained with varying pretraining data.

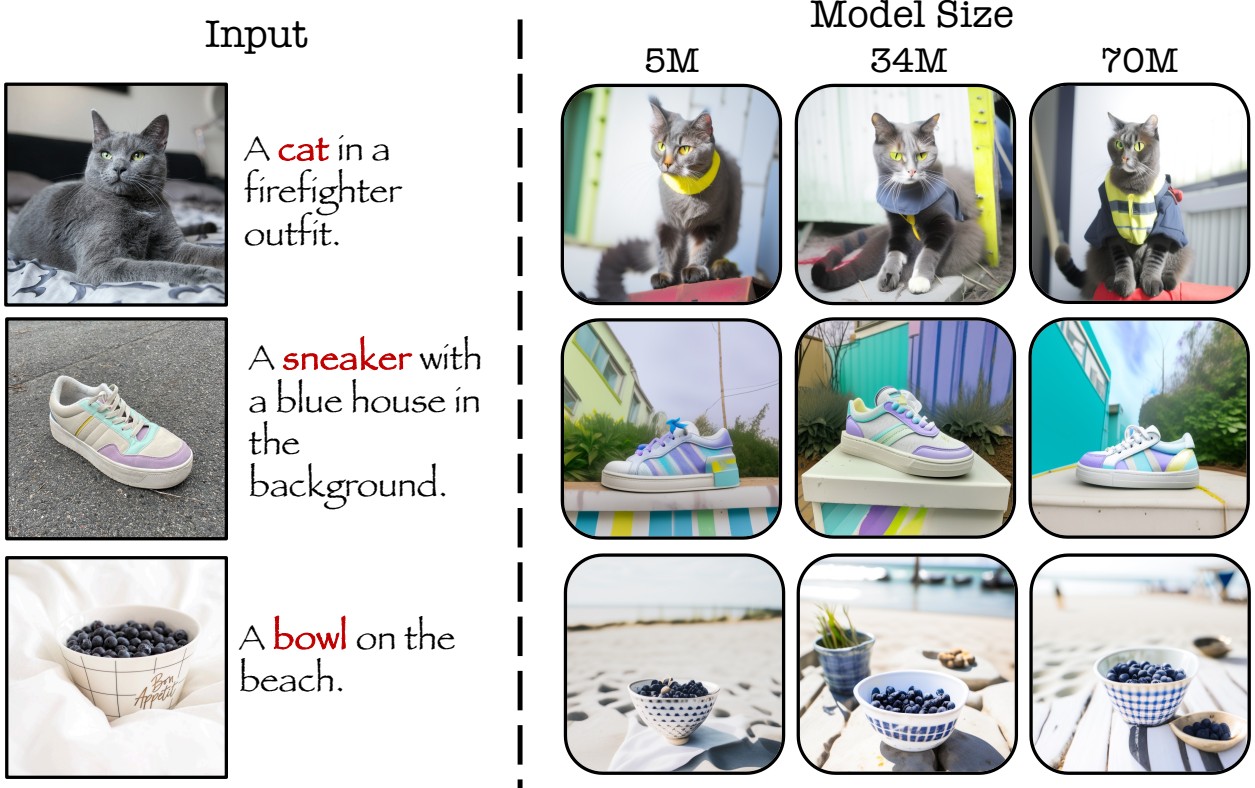

Figure 13: Qualitative examples $\lambda$-*ECLIPSE* model trained with varying model parameters.

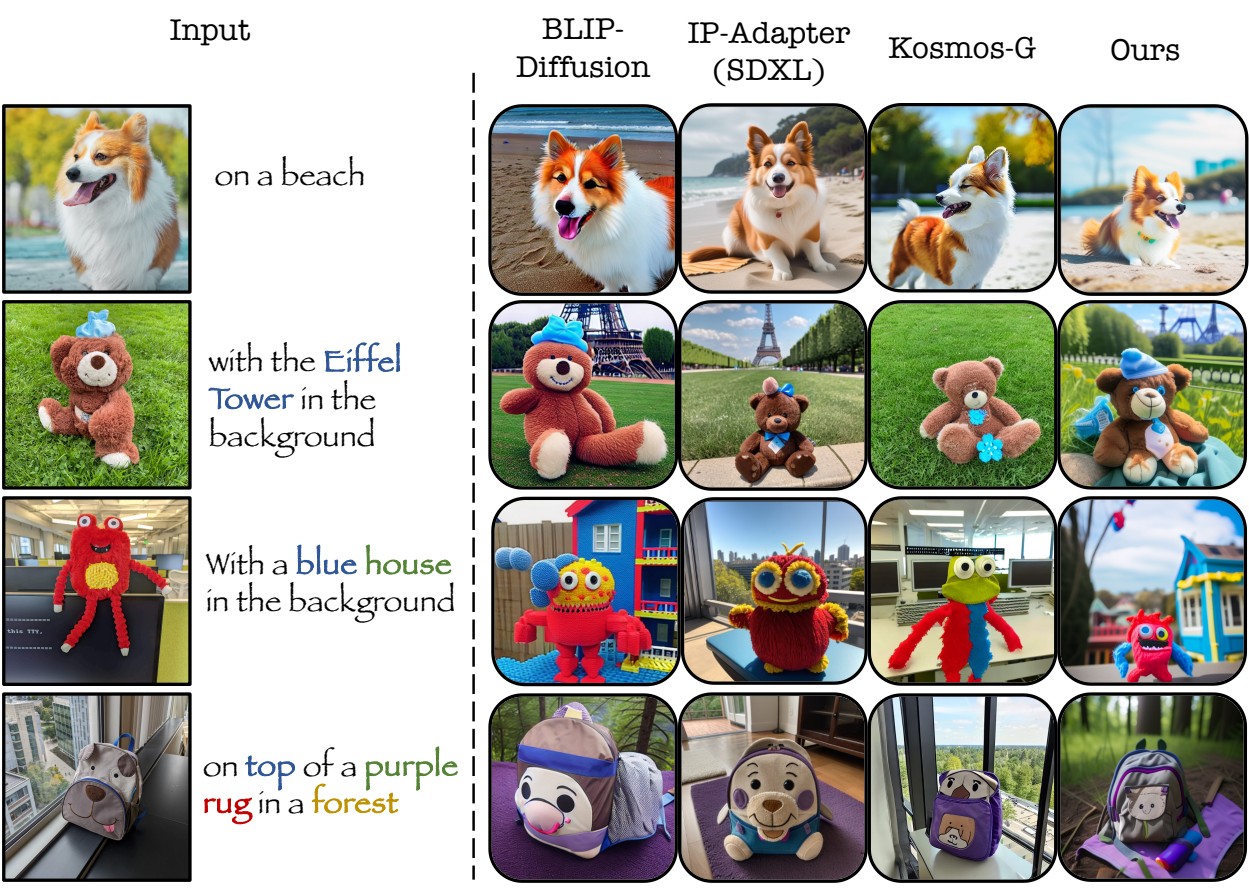

Figure 14: Qualitative examples of the increasing complexity of novel visual concepts as we move from top to bottom.

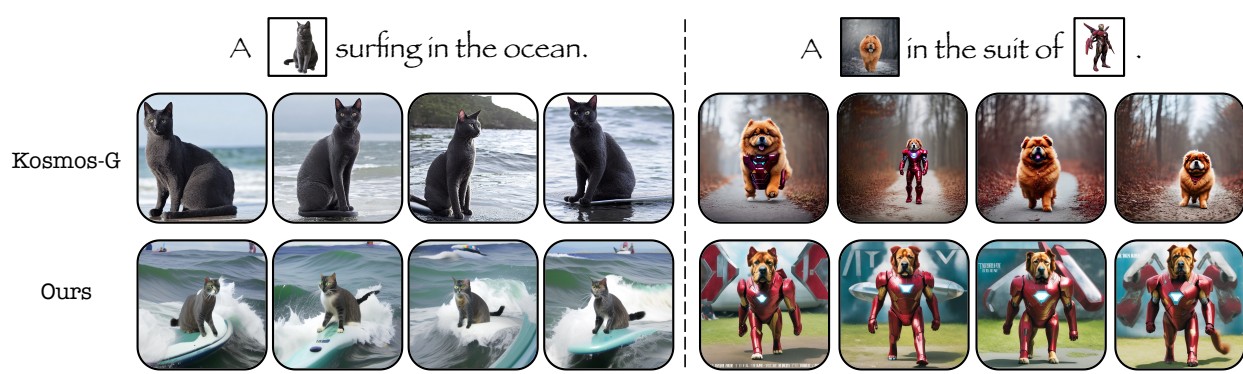

Figure 15: Qualitative examples of showcasing the consistency comparisons between Kosmos-G and λ-ECLIPSE.

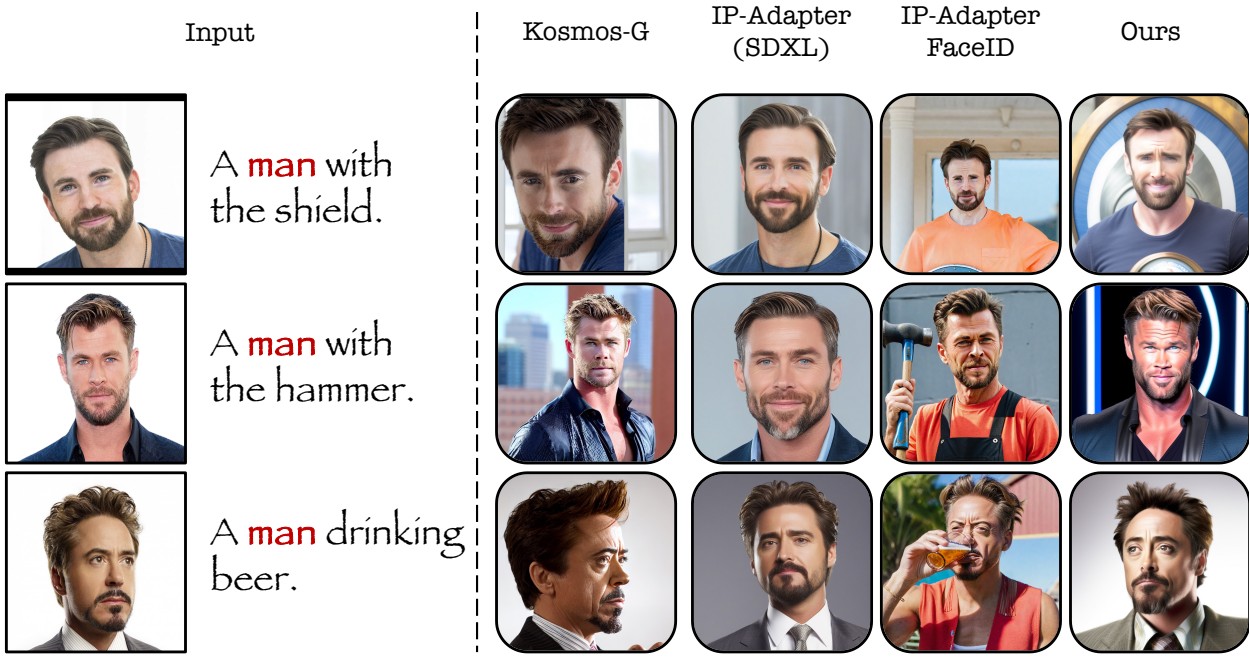

Figure 16: Qualitative examples of showcasing the failure cases on human faces on Kosmos-G, IP-Adapter (SDXL), IP-Adapter (FaceID), and $\lambda$-*ECLIPSE*.

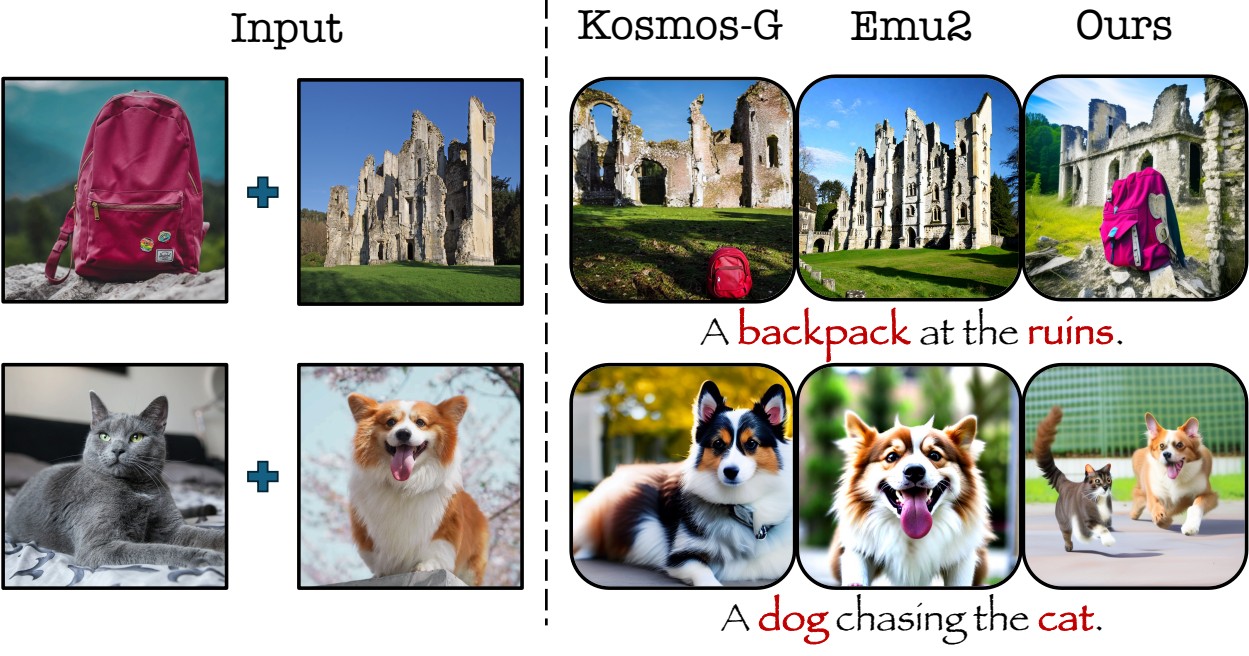

Figure 17: Qualitative examples of showcasing the failure cases on Multibench of Kosmos-G, Emu2, and $\lambda$-*ECLIPSE*.

