# OpenReview forum: "λ-ECLIPSE: Multi-Concept Personalized Text-to-Image Diffusion Models by Leveraging CLIP Latent Space"
_TMLR — Accepted by TMLR_

### Review · Reviewer_4XuP · 2024-07-25

**Summary Of Contributions:**

This work proposes a simple and efficient method for finetuning-free Multi-Concept Personalized Text-to-Image with only 34M parameters. The main idea is to fully use the powerful CLIP latent space. Experiment results show that the proposed method can balance the edge map, subjects, and composition well and achieve comparable performance with much less computation resources.

**Audience:**

Yes

**Broader Impact Concerns:**

The broader impact has been discussed.

**Claims And Evidence:**

Yes

**Requested Changes:**

The explanations on the coarse alignment between the edge map and generated image.
Experiments on diffusion models.

**Strengths And Weaknesses:**

Pros:

The proposed λ-ECLIPSE requires much fewer parameters and computation resources to achieve comparable performance compared to prior subject-driven image generation methods.

The proposed method proposes to use an interleaved feature representation and trains a lightweight projector (34M parameters with 74 GPU hours) to transform the interleaved features into an aligned embedding space. The main image generation backbone is frozen.

To equip spatial control, the edge map condition is also involved during training with dropout. Coarse spatial control can be enabled with id-preservation.

Due to the design of interleaved feature representation, multi-subject P-T2I and concept interpolation can be achieved naturally.

Abundant experiments and ablations have been performed to demonstrate the superiority of the proposed method on the concept and composition alignment.

Cons:

As to the claim of diffusion-free in Table 1, I think it is just a learning paradigm instead of an advantage. Maybe something like model-agnostic is a better choice if the authors want to claim the generalization of the proposed method. BTW, the title of this work is “Multi-Concept Personalized Text-to-Image Diffusion Models”

It is claimed that the prior methods do not preserve the identity well when using edge map control. If I understand correctly, the proposed λ-ECLIPSE uses the edge maps as an extra vision embedding in the interleaved features.
While the edge map in Figure 3 does not align well with the image in the re-generated columns, I wonder if the control here is just a coarse-level control for something like layout or sketch.

I wonder if the proposed technique can also be applied to the diffusion models like StabeDiffusion. If so, experiments on this can make this work more comprehensive.

---

> ### Author Response · Authors · 2024-08-15
> **Response to Reviewer 4XuP**
>
> We greatly appreciate your thorough review and are encouraged by your positive feedback. We are pleased that you recognize λ-ECLIPSE as a method that requires **“much fewer parameters and computation resources,”** offers **“coarse spatial control,”** and is supported by **“abundant experiments and ablations.”**
>
> Below, we provide additional clarifications as requested:
>
> > ### Generalization to Other T2I Diffusion Models
>
> Our approach is designed to generalize across any pretrained UnCLIP diffusion models, including Stable-UnCLIP/SDv2.1, Karlo, and Kandinsky v2.2. We conducted additional pretraining experiments to substantiate this claim and demonstrated the generalization ability of λ-ECLIPSE.
>
> We would like to reiterate the core pipeline of λ-ECLIPSE or UnCLIP models, as detailed in Appendix A. The UnCLIP model comprises two key modules: (1) the Prior model, which maps the text embedding to the image embedding, and (2) the Diffusion rendering model, which synthesizes the image based on the estimated image embeddings from the prior model. For our generalization studies, we adhered to two essential criteria:
>
> - Selection of an open-source UnCLIP (2 stage) model, where we chose Stable-UnCLIP, a modified version of Stable Diffusion v2.1, to accept image embeddings as input.
> - Choice of pretrained CLIP model, with Stable-UnCLIP utilizing the OpenAI-CLIP-ViT-L/14 model. Accordingly, we trained λ-ECLIPSE with this model as a preliminary feature extractor.
>
> Following these guidelines, we trained λ-ECLIPSE with the same parameters and evaluated its performance on DreamBench. Our results confirm that λ-ECLIPSE effectively generalizes to different diffusion models and CLIP variants. Notably, the Stable-UnCLIP variant of λ-ECLIPSE achieves performance similar to Emu2, reinforcing the superiority of our initial design choices.
>
> | **Model**                                 | **DINO** | **CLIP-I** | **CLIP-T** |
> |:------------------------------------------|:-------:|:---------:|:---------:|
> | **Stable UnCLIP (Stable Diffusion v2.1)** | 0.564   | 0.778     | 0.276 |
> | **Kandinsky v2.2**                        | **0.613** | **0.783** | **0.307**    |
>
> **Changes in the paper:** We have included this ablation study in Appendix G.1 and Table 10.
>
>
> > ### Additional Minor Clarifications
>
> **[W1] Improving the clarification behind “diffusion-free” keyword**
>
> Thank you for highlighting the importance of precise wording. We agree that describing λ-ECLIPSE as a training-time “diffusion-agnostic” approach enhances clarity for future readers, and we have revised the draft accordingly.
>
> Moreover, while λ-ECLIPSE introduces a novel paradigm for training P-T2I models in a diffusion-agnostic manner, we want to emphasize that our contribution goes beyond this. Our work demonstrates that pretrained self-supervised embeddings from a CLIP-like model are sufficiently expressive to perform P-T2I-like tasks — an insight previously unknown to the community. This foundational advance naturally leads to significant resource efficiency as a byproduct.
>
> **[W2] Coarse-Level Control via Canny-Edge Maps**
>
> Indeed, λ-ECLIPSE utilizes Canny-edge maps to achieve coarse-level control, as illustrated in Figure 3. We observed that while CLIP struggles to capture the exact outlines of edge-map images, this characteristic enables the coarse-level control that our method provides.
>
> In Table 5, we also compare our approach with baselines that integrate pretrained ControlNet [1] with Canny-edge maps. **These methods involve additional finetuning similar to λ-ECLIPSE** and, as reported, result in suboptimal outcomes, particularly in terms of personalization. In contrast, the coarse-level control achieved by λ-ECLIPSE is notably more stable than that of the baselines.
>
> Having said that, we fully agree that clarifying λ-ECLIPSE’s provision of “coarse-level” control rather than hard controls will enhance the clarity and impact of our work. **This distinction underscores the significance of λ-ECLIPSE, particularly compared to resource-intensive systems like ControlNet, for achieving stable coarse-level control.**
>
>
> We trust that these clarifications address all your concerns and questions. We look forward to further discussion if needed.
>
> ---
>
> [1] Zhang, Lvmin, Anyi Rao, and Maneesh Agrawala. "Adding conditional control to text-to-image diffusion models." In Proceedings of the IEEE/CVF International Conference on Computer Vision, pp. 3836-3847. 2023.

---

### Review · Reviewer_fKg9 · 2024-08-04

**Summary Of Contributions:**

This paper studies subject driven personalized/customized text-to-image generation, and in comparison to related works in this literature, specifically tries to answer this question: Do we really need diffusion models to train the customization models?
Through empirical evidences, they find that CLIP vision latent space is already expressive enough to preseve fine-grained details. They leverage this finding and propose $\lambda$-Eclipse, based on the properties of UnCLIP text-to-image models. $\lambda$-Eclipse aligns the output with CLIP vision space instead of the CLIP text space in LDMs. They claim to be the first to offer multi-concept-driven generation without depending on diffusion UNet models (except for inference). Their method is includes light finetuning using roughly small dataset, but high quality pairs of text and image that can achieve competitive results.

**Audience:**

Yes

**Broader Impact Concerns:**

- The method is heavily based on pre-trained CLIP models, and hence may be highly impacted by their weaknesses, such as incorporating NSFW contents. [1]


[1] Kazemi, Hamid, et al. "What do we learn from inverting CLIP models?." arXiv preprint arXiv:2403.02580 (2024).

**Claims And Evidence:**

Yes

**Requested Changes:**

already discussed in weaknesses

**Strengths And Weaknesses:**

Stregths:
+ Method is well-defined and described. Text is well written and easy to follow.
+ Related works are well covered including their drawbacks. Comparison to related works gives a good picture.
+ While proposed method is light weight it achieves competitive results. Being resource efficient (both is parameter space, and the need for data) is a huge plus.
+ Including additional modality such as Canny edge map, that provides more refined control over subject-driven text-to-image generation.
+ Evaluation of the individual contributions of different components in $\lambda$-ECLIPSE loss function.

Weaknesses:
- Fig4: How does proposed generatin+ selection differs if we generate 100 images instead of 4? How much of variation one may expect from different models, including proposed method?
- Any results that support this statement in 3.2? "Preliminary experiments indicated that this method does not effectively capture the intricate relationships between target text tokens (e.g. “dog”) and the corresponding concept images"
- What kind of filters are used when sampling 2M from LAION? resolution/aesthetic, etc?
- How did you find the quality of captions generated by BLIP2? How does using stronger captioning models/VLMs? e.g. CogVLM, or MiniCPM would affect the quality of results?
- Why did you choose Kv.2.2 ober other UNet models such as SDXL?
- as discussed in limitations, what is missing is the effect of number of parameters and dataset size on the quality of results.

---

> ### Author Response · Authors · 2024-08-15
> **Response to Reviewer fKg9 (part 1)**
>
> We are greatly encouraged by your review and appreciate your comprehensive evaluation of our paper. We are delighted that you found λ-ECLIPSE to be **well-defined, well-written, and supported by comprehensive related work and detailed ablations**. We are especially pleased that you recognize our work as **“lightweight and achieving competitive results,”** and acknowledge the significant advantage of its resource efficiency in both parameter space and data requirements.
>
> Please find our clarifications below:
>
> > ### Variations among the Synthesized Images [W1]
>
> We fully agree that low variation in synthesized images is crucial. As shown in Figure 15 of Appendix H, λ-ECLIPSE consistently produces stable results across different random seeds, reinforcing the robustness of our approach.
>
> Moreover, we refer you to the performance comparison on the ConceptBed [1] benchmark (Table 3). ConceptBed introduces the “Concept Replication” metric, which specifically accounts for variations in generated images. Following this benchmark, we synthesized 100 images per concept (totaling 80 unique concepts) and reported the Concept Confidence Deviation (CCD) metric. As evidenced in Table 3, zero-shot λ-ECLIPSE outperforms all finetuning-free baselines, indicating the lowest variations concerning the reference concepts among all finetuning-free methods.
>
> > ### Ablation on Interleaved Pretraining [W2]
>
> We conducted additional experiments to support our statement from Section 3.2 by training the model without interleaved data. Specifically, we concatenated the prompt embedding from the CLIP text encoder with the concept-specific image embedding and trained the model with identical hyperparameters. The performance comparisons on DreamBench (single concept) and Multibench (multiple concepts) are shown below. Our findings indicate that without interleaved data, the model's ability to align concepts decreases as the number of concepts increases, resulting in a sharp 6% drop in the DINO score.
>
> |  **# of Subjects** | | **Metric** | **With Interleaved** | **Without Interleaved** | **Difference** |
> |:----:|:----:|:-:|:--:|:----:|:-----:|
> | **1** | | **DINO**   | 0.613  | 0.6006   | **-2.02%**     |
> | | | **CLIP-I** | 0.783    | 0.7882     | 0.66%      |
> | | | **CLIP-T** | 0.307  | 0.3048   | **-0.72%**     |
> | **2** | | **DINO**   | 0.4478 | 0.4332   | **-3.26%**     |
> | | | **CLIP-I** | 0.7409   | 0.7384     | **-0.34%**     |
> | | | **CLIP-T** | 0.3327 | 0.3271   | **-1.68%**     |
> | **3** | | **DINO**   | 0.342  | 0.3202   | **-6.37%**     |
> | | | **CLIP-I** | 0.6463   | 0.648     | 0.26%      |
> | | | **CLIP-T** | 0.3469 | 0.3469   | 0.00%      |
>
> This performance degradation is primarily due to attribute leakage. For example, when the model is tasked with generating “A backpack at the ruins” + <backpack> + <ruins>, the non-interleaved pretrained models tend to generate the backpack with the color of the ruins. We have provided qualitative examples in the appendix (Figure 11).
>
> **Changes in the paper:** We added this ablation study and qualitative results in Appendix G.2, Table 11 and Figure 11.
>
> > ### Applied Filters on LAION [W3]
>
> We have detailed the applied filters in Appendix B.
>
> > ### Choice of VLMs [W4]
>
> The choice of Vision-Language Models (VLMs) is indeed critical in training T2I models on synthetic datasets, as noted in prior works [2,3]. However, in our case, we deliberately avoided stronger VLMs for two key reasons:
>
> - Stronger VLMs often provide excessive detail, sometimes hallucinating information, making them difficult to control with simple instructions. BLIP2, on the other hand, offers shorter but more precise captions.
> - For example, with a training prompt like “A black <cat> wearing blue <sunglasses>,” where <.> represents the image embedding, a stronger VLM might spuriously correlate the target image with the “black cat” rather than focusing on the appearance of the entities from the reference image.
>
> Therefore, we prioritized precision in our VLM choice, opting for BLIP2, which aligns perfectly with our needs for concise and accurate captions.

---

> ### Author Response · Authors · 2024-08-15
> **Response to Reviewer fKg9 (part 2)**
>
> > ### Choice of UNet Model [W5]
>
> Any UNet model capable of accepting CLIP vision embeddings as input could theoretically be utilized. Kandinsky v2.2, being the SOTA model that accepts CLIP-ViT-bigG vision embeddings, is a natural choice. Therefore, our approach is designed to generalize across any pretrained UnCLIP diffusion models, including Stable-UnCLIP/SDv2.1, Karlo, and Kandinsky v2.2.
>
> We conducted additional pretraining experiments to substantiate this claim and demonstrated the generalization ability of λ-ECLIPSE to Stable-UnCLIP (a.k.a. Stable Diffusion v2.1). We trained λ-ECLIPSE with the same parameters and evaluated its performance on DreamBench. Our results confirm that λ-ECLIPSE effectively generalizes to diffusion models and CLIP variants. Notably, the Stable-UnCLIP variant of λ-ECLIPSE achieves performance similar to Emu2, reinforcing the superiority of our initial design choices.
>
> | **Model**                                 | **DINO** | **CLIP-I** | **CLIP-T** |
> |:------------------------------------------|:-------:|:---------:|:---------:|
> | **Stable UnCLIP (Stable Diffusion v2.1)** | 0.564   | 0.778     | 0.276 |
> | **Kandinsky v2.2**                        | **0.613** | **0.783** | **0.307**     |
>
>
> **Changes in the paper:** We have included this ablation study in Appendix G.1 and Table 10.
>
> > ### Ablations on Parameter and Data Sizes [W6]
>
> As noted in the limitations, this was not the primary focus of our work. Nonetheless, we have demonstrated that our model achieves the intended goals within the current settings. To enhance the comprehensiveness of our work, we conducted extensive ablations on varying data sizes (100k, 500k, 1M, and 2M) and model parameter sizes (5M, 35M, and 70M). The table below reports the performance of these newly trained models on DreamBench. Due to the rebuttal's time constraints, all models were trained with identical hyperparameters as the proposed λ-ECLIPSE, though this may not fully optimize the larger models (e.g., 70M parameters).
>
> | **Data Size** | **DINO** | **CLIP-I** | **CLIP-T** |
> |:-------------|:-------:|:---------:|:---------:|
> | **100k**      | 0.593   | **0.786** | 0.301     |
> | **500k**      | 0.595   | 0.777     | 0.305     |
> | **1M**        | 0.596   | 0.778     | 0.306     |
> | **2M**        | **0.613**   | 0.783     | **0.307** |
>
>
> | **Model Size** | **DINO** | **CLIP-I** | **CLIP-T** |
> |:--------------|:-------:|:---------:|:---------:|
> | **5M**         | 0.586   | 0.779     | 0.305     |
> | **34M**        | **0.613**   | **0.783**     | 0.307 |
> | **70M**        | 0.593   | 0.775     | **0.309**     |
>
> The results show that data size influences model performance, with larger datasets improving concept understanding and prompt compositions. Notably, λ-ECLIPSE with only 5M parameters delivers performance close to that of larger models, and the 34M parameter model even surpasses the 70M parameter model in terms of DINO score. However, qualitative results (Figure 13) show that increasing model parameters enhances qualitative performance and concept alignment. Specifically, the 70M parameter model excels in generating finer details of the reference concept while adhering closely to text prompts. The 34M model offers a more balanced trade-off between performance and resource efficiency.
>
> **Changes in the paper:** Given the importance of the model parameter size ablation, we have included these results in Appendix G.3, along with Tables 12-13 and Figures 12-13.
>
> We believe our response addresses your concerns comprehensively and encourage you to reevaluate our submission. We look forward to further discussion.
>
> ---
>
> [1] Patel, Maitreya, Tejas Gokhale, Chitta Baral, and Yezhou Yang. "Conceptbed: Evaluating concept learning abilities of text-to-image diffusion models." In Proceedings of the AAAI Conference on Artificial Intelligence, vol. 38, no. 13, pp. 14554-14562. 2024.
>
> [2] Betker, James, Gabriel Goh, Li Jing, Tim Brooks, Jianfeng Wang, Linjie Li, Long Ouyang et al. "Improving image generation with better captions." Computer Science. https://cdn. openai. com/papers/dall-e-3. pdf 2, no. 3 (2023): 8.
>
> [3] Chen, Junsong, Chongjian Ge, Enze Xie, Yue Wu, Lewei Yao, Xiaozhe Ren, Zhongdao Wang, Ping Luo, Huchuan Lu, and Zhenguo Li. "Pixart-\sigma: Weak-to-strong training of diffusion transformer for 4k text-to-image generation." arXiv preprint arXiv:2403.04692 (2024).

---

### Review · Reviewer_akk3 · 2024-08-06

**Summary Of Contributions:**

The paper proposed a new method for Personalized Text-to-Image (P-T2I) called $\lambda$-ECLIPSE. $\lambda$-ECLIPSE shares similar idea as UnCLIP, which generates the CLIP image embedding from the text description before mapping it to the image via the diffusion model. $\lambda$-ECLIPSE takes multiple images and text instructions as input and estimates the respective vision embedding. The vision embedding is fed to a diffusion UNet to generate the image. Compared with other P-T2I methods, $\lambda$-ECLIPSE is finetuning-free and is compatible with the pretrained diffusion model. It also has comparable performance with state-of-the-art P-T2I methods.

**Audience:**

Yes

**Broader Impact Concerns:**

No.

**Claims And Evidence:**

Yes

**Requested Changes:**

The author needs to justify whether $\lambda$-ECLIPSE is generalizable to other text-image pretrained models, and conduct ablation study about the impact of interleaved pretraining.

**Strengths And Weaknesses:**

Strengths:

$\lambda$-ECLIPSE is efficient at test-time and requires less resources to pretrain than Kosmos-G. As shown in Table 2, $\lambda$-ECLIPSE needs 0.2 GPU hours to finetune while Textual Inversion needs 1 GPU hour. The fast speed is due to its algorithm design. $\lambda$-ECLIPSE trains a mapping function that encapsulates both text prompts and subject visuals into an image embedding, which is further adopted to generate image with a frozen-weight diffusion model. Since the diffusion model won't be updated, the majority of the training cost comes from training the mapping function. The author proposed to combine the concept alignment loss function and the contrastive loss function in training the mapping function, and also pretrain it on interleaved text-image dataset.

Apart from being efficient, the performance of $\lambda$-ECLIPSE is comparable to baselines on Dreambench and ConceptBed, according to Table 2 and Table 3.


Weaknesses:

The author conducted ablation study on some design choices of $\lambda$-ECLIPSE, including the combination of projection loss (or concept alignment loss) and contrastive loss. However, I haven't seen ablation study on interleaved pretraining. How important is interleaved pretraining for $\lambda$-ECLIPSE? In addition,  $\lambda$-ECLIPSE is the only method that relies on Kv2.2 according to Table 7. This casts some doubts on how well the model generalizes to other pretrained text-image diffusion models. Another shortcoming is that the method underperforms BLIP-DIffusion and Textual Inversion in concept alignment. This may indicate that the algorithm is prone to hallucination.

---

> ### Author Response · Authors · 2024-08-15
> **Response to Reviewer akk3 (part 1)**
>
> Thank you for your thoughtful review of our paper, λ-ECLIPSE. We are pleased that you recognized our work as **“efficient at test-time and requires fewer resources”** while achieving competitive performance.
>
> However, we would like to emphasize that our contribution goes beyond resource efficiency in P-T2I tasks. Our key insight is demonstrating that pretrained self-supervised embeddings from a CLIP-like model are expressive enough to perform P-T2I-like tasks — a novel finding in the community. **Resource efficiency, therefore, is a natural byproduct of this fundamental insight.**
> Please find our detailed responses to your specific inquiries below:
>
> Please find our detailed responses to your specific inquiries below:
>
>
> > ### Generalization to Other T2I Diffusion Models
>
> Our approach is designed to generalize across any pretrained UnCLIP diffusion models, including Stable-UnCLIP/SDv2.1, Karlo, and Kandinsky v2.2. We conducted additional pretraining experiments to substantiate this claim and demonstrated the generalization ability of λ-ECLIPSE.
>
> We would like to reiterate the core pipeline of λ-ECLIPSE or UnCLIP models, as detailed in Appendix A. The UnCLIP model comprises two key modules: (1) the Prior model, which maps the text embedding to the image embedding, and (2) the Diffusion rendering model, which synthesizes the image based on the estimated image embeddings from the prior model. For our generalization studies, we adhered to two essential criteria:
>
> - Selection of an open-source UnCLIP (2 stage) model, where we chose Stable-UnCLIP, a modified version of Stable Diffusion v2.1, to accept image embeddings as input.
> - Choice of pretrained CLIP model, with Stable-UnCLIP utilizing the OpenAI-CLIP-ViT-L/14 model. Accordingly, we trained λ-ECLIPSE with this model as a preliminary feature extractor.
>
> Following these guidelines, we trained λ-ECLIPSE with the same parameters and evaluated its performance on DreamBench. **Our results confirm that λ-ECLIPSE effectively generalizes to different diffusion models and CLIP variants**. Notably, the Stable-UnCLIP variant of λ-ECLIPSE achieves performance similar to Emu2, **reinforcing the superiority of our initial design choices**.
>
> | **Model**                                 | **DINO** | **CLIP-I** | **CLIP-T** |
> |:------------------------------------------|:-------:|:---------:|:---------:|
> | **Stable UnCLIP (Stable Diffusion v2.1)** | 0.564   | 0.778     | 0.276 |
> | **Kandinsky v2.2**                        | **0.613** | **0.783** | **0.307**     |
>
> **Changes in the paper:** We have included this ablation study in Appendix G.1 and Table 10.
>
> > ### Ablation on Interleaved Pretraining
>
> As mentioned in Section 3.2, we find that training λ-ECLIPSE without interleaved pretraining would yield similar results for “single-concept” P-T2I tasks. However, for “multi-concept” personalization, our experiments reveal that models trained without interleaved data sometimes struggle to synthesize the desired images.
>
> In response to your request, we conducted additional experiments by training the model without interleaved data. Specifically, we concatenated the prompt embedding from the CLIP text encoder with the concept-specific image embedding and trained the model with identical hyperparameters. The performance comparisons on DreamBench (single concept) and Multibench (multiple concepts) are shown below. Our findings indicate that without interleaved data, the model's ability to align concepts decreases as the number of concepts increases, resulting in a sharp 6% drop in the DINO score.
>
> | **# of Subjects** |  | **Metric** | **With Interleaved** | **Without Interleaved** | **Difference** |
> |:----:|:----:|:-:|:--:|:----:|:-----:|
> | **1** | | **DINO**   | 0.613  | 0.6006   | **-2.02%**     |
> | | | **CLIP-I** | 0.783    | 0.7882     | 0.66%      |
> | | | **CLIP-T** | 0.307  | 0.3048   | **-0.72%**     |
> | **2** | | **DINO**   | 0.4478 | 0.4332   | **-3.26%**     |
> | | | **CLIP-I** | 0.7409   | 0.7384     | **-0.34%**     |
> | | | **CLIP-T** | 0.3327 | 0.3271   | **-1.68%**     |
> | **3** | | **DINO**   | 0.342  | 0.3202   | **-6.37%**     |
> | | | **CLIP-I** | 0.6463   | 0.648      | 0.26%      |
> | | | **CLIP-T** | 0.3469 | 0.3469   | 0.00%      |
>
>
>
> This performance degradation is primarily due to attribute leakage. For example, when the model is tasked with generating “A backpack at the ruins” + <backpack> + <ruins>, the non-interleaved pretrained models tend to generate the backpack with the color of the ruins. We have provided qualitative examples in the appendix (Figure 11).
>
> **Changes in the paper:** We added this ablation study and qualitative results in Appendix G.2, Table 11, and Figure 11.

---

> > ### Author Response · Authors · 2024-08-15
> > **Response to Reviewer akk3 (part 2)**
> >
> > > ### Clarification on Concept Alignment
> >
> > We would like to clarify the metrics used in ConceptBed [1]: (1) “Concept Replication” measures concept overfitting without composition changes in the prompt, (2) “Composition Alignment” assesses image-text alignment without considering concept similarity, and (3) “Concept Alignment” evaluates concept replication with diverse changes in text prompts.
> >
> > An ideally balanced system should maintain SOTA performance on Concept Replication and Composition Alignment while achieving a strong but balanced score on Concept Alignment. A low Concept Alignment score can indicate that the model prioritizes replicating the concept image over aligning it with the text prompt and vice-versa. The other two metrics further validate this. When we average the performances, λ-ECLIPSE consistently outperforms other finetuning-free approaches.
> >
> > | **Method**           | **Concept Replication (↓)** | **Concept Alignment (↓)** | **Composition Alignment (↓)** | **Average (↓)** |
> > |:---------------------|:--------------------------:|:-------------------------:|:-----------------------------:|:---------------:|
> > | **ELITE**             | 0.3195                     | 0.4666                    | 0.1832                        | 0.3231          |
> > | **BLIP-Diffusion**    | 0.3510                     | **0.3245**                    | 0.1589                        | 0.2781          |
> > | **IP-Adapter**        | 0.3665                     | 0.3571                    | 0.0641                        | 0.2626          |
> > | **λ-ECLIPSE**         | **0.2853**                     | 0.3619                    | **-0.0200**                      | **0.2091**          |
> >
> > **Changes in the paper:** We added an average column in Table 3.
> >
> > We trust that our response addresses your concerns comprehensively and encourage you to reevaluate our submission positively. We look forward to further discussion.
> >
> > ---
> >
> > [1] Patel, Maitreya, Tejas Gokhale, Chitta Baral, and Yezhou Yang. "Conceptbed: Evaluating concept learning abilities of text-to-image diffusion models." In Proceedings of the AAAI Conference on Artificial Intelligence, vol. 38, no. 13, pp. 14554-14562. 2024.

---

### Author Response · Authors · 2024-08-15
**Global Response to Reviewers**

We sincerely appreciate the constructive feedback provided by the reviewers and AE for managing the process. It is gratifying to observe the positive evaluations across various dimensions of our work, as highlighted unanimously by the reviewers.

- The reviewers consistently recognize our paper as **“the most efficient approach”** with **“competitive performance”** (Reviewers akk3, fKg9, 4XuP).
- Reviewer 4XuP noted that **“abundant experiments and ablations”** have been performed to demonstrate the superiority of the proposed method.
- Reviewers fKg9 and 4XuP also emphasized that our approach provides **“better coarse-level control”** for P-T2I tasks and facilitates **“multi-subject interpolations naturally,”** respectively.
- Additionally, we are pleased that Reviewer fKg9 found our method and paper to be **“well-defined and easy to follow.”**

We have provided detailed responses to each reviewer individually. Below, we summarize our responses to key questions and provide additional clarifications. Furthermore, **we have revised the paper** to include all the experiments conducted during the rebuttal phase, along with additional qualitative results.

> ### Important Clarification Regarding UnCLIP vs. Stable Diffusion

UnCLIP (a.k.a. DALL-E 2 [1]) and Stable Diffusion (a.k.a. LDM [2]) models represent two parallel families of T2I models that differ in a crucial aspect:
- Stable Diffusion models are conditioned to render images using only “text prompts.”
- In contrast, UnCLIP models follow a dual approach: first, estimating the image embedding conditioned on the “text prompts” and second, using a UNet-based image rendering model conditioned on the “image embeddings.” We further elaborate on UnCLIP models in Appendix A.

In this work, we leverage the “image-embedding” conditioning of UnCLIP models to render images. This capability allows us to demonstrate that the CLIP latent space, though not explicitly trained to preserve fine-grained image details, is sufficient for performing P-T2I tasks, leading to significant resource efficiency.

Our method primarily relies on Kandinsky v2.2, the SOTA open-source UnCLIP model. However, it is generalizable to any diffusion model that accepts image embeddings as additional conditions. We demonstrate this by extending λ-ECLIPSE to Stable-UnCLIP (i.e., Stable Diffusion v2.1).

> ### Summary of Additional Ablations

- **Ablations on Data and Model Sizes:** As requested by Reviewer fKg9, we performed additional ablations by varying data sizes (100k, 500k, 1M, and 2M) and model sizes (5M, 34M, and 70M). We observed that larger data sizes directly correlate with improved performance of λ-ECLIPSE. Notably, varying model sizes enhances the balance between concept and composition alignment, particularly in qualitative comparisons.
- **Ablation on Interleaved Pretraining:** In response to Reviewer akk3’s request, we conducted comparisons on training λ-ECLIPSE with and without interleaved data. Our findings indicate that training without interleaved data degrades performance as the number of subjects increases, with a 6% drop in the DINO score for three-subject P-T2I on the Multibench dataset.

> ### Changes in the Paper:

- We have included all the new ablation studies in Appendix G, along with tables containing quantitative results and figures showcasing qualitative comparisons.
- Additionally, we revised two keywords to improve clarity as suggested by Reviewer 4XuP:
    - “diffusion-free” → “training-time diffusion-agnostic”
    - “Canny-edge map controls” → “Canny-edge based coarse-level controls”

Once again, we thank the reviewers and the Action Editor for their time and detailed feedback. We hope that our responses have addressed all remaining concerns and questions. We look forward to further discussion if needed.

---

[1] Ramesh, Aditya, Prafulla Dhariwal, Alex Nichol, Casey Chu, and Mark Chen. "Hierarchical text-conditional image generation with clip latents." arXiv preprint arXiv:2204.06125 1, no. 2 (2022): 3.

[2] Rombach, Robin, Andreas Blattmann, Dominik Lorenz, Patrick Esser, and Björn Ommer. "High-resolution image synthesis with latent diffusion models." In Proceedings of the IEEE/CVF conference on computer vision and pattern recognition, pp. 10684-10695. 2022.

---

### Decision · Action_Editor_DtpZ · 2024-10-29

**Recommendation:** Accept as is

**Comment:**

The core findings and contributions of the paper is that the CLIP vision latent contains sufficiently expressive appearance details. This leads to the proposed prior-training strategy without resorting to finetuning diffusion models for personalization.

The initial reviews recognized the interesting idea and the solid set of experiments. The rebuttal further provides additional ablation on data/model sizes and interleaved pretraining.

After the rebuttal, all reviewers are satisfied with the rebuttal and are leaning to accept the paper.

**Audience:**

Yes, I believe the paper will attract audience in the CV/ML community.

**Claims And Evidence:**

The paper presents a finetuning-free subject-driven test-to-image generation model. The claims made in the submission are supported by convincing evidence. In particular, it demonstrates comparable results while bing more efficient and requiring fewer resources. Additional results further showcase competitive performance with respect to the state-of-the-art on ConceptBed (Table 3), Multibench (Table 4), and Dreambench (Table 5). The reviewers all agree that the experimental validations are sufficient to support the claims.